# Optogenetic modulation of TDP-43 oligomerization accelerates ALS-related pathologies in the spinal motor neurons

Kazuhide Asakawa [1,2,3✉], Hiroshi Handa [1] & Koichi Kawakami [2,3✉]

Cytoplasmic aggregation of TDP-43 characterizes degenerating neurons in most cases of amyotrophic lateral sclerosis (ALS). Here, we develop an optogenetic TDP-43 variant (opTDP-43), whose multimerization status can be modulated in vivo through external light illumination. Using the translucent zebrafish neuromuscular system, we demonstrate that short-term light stimulation reversibly induces cytoplasmic opTDP-43 mislocalization, but not aggregation, in the spinal motor neuron, leading to an axon outgrowth defect associated with myofiber denervation. In contrast, opTDP-43 forms pathological aggregates in the cytoplasm after longer-term illumination and seeds non-optogenetic TDP-43 aggregation. Furthermore, we find that an ALS-linked mutation in the intrinsically disordered region (IDR) exacerbates the light-dependent opTDP-43 toxicity on locomotor behavior. Together, our results propose that IDR-mediated TDP-43 oligomerization triggers both acute and long-term pathologies of motor neurons, which may be relevant to the pathogenesis and progression of ALS.

[1] Department of Chemical Biology, Tokyo Medical University, Shinjuku-ku, Tokyo 160-8402, Japan. [2] Division of Molecular and Developmental Biology, National Institute of Genetics, 1111 Yata, Mishima, Shizuoka 411-8540, Japan. [3] Department of Genetics, Graduate University for Advanced Studies (SOKENDAI), 1111 Yata, Mishima, Shizuoka 411-8540, Japan. ✉email: kasakawa@nig.ac.jp; kokawaka@nig.ac.jp

Amyotrophic lateral sclerosis (ALS) is a neurological disorder in which the upper and lower motor neurons progressively degenerate, leading to muscular atrophy and eventually fatal paralysis. Trans-activation response element (TAR) DNA-binding protein 43 (TDP-43), a heterogeneous nuclear ribonucleoprotein, is mislocalized to the cytoplasm and forms pathological aggregates in the degenerating motor neurons in ALS[1,2]. TDP-43 aggregation characterizes almost all cases of sporadic ALS[3,4], which accounts for greater than 90% of ALS. Moreover, mutations in the *TARDBP* gene encoding TDP-43 are linked to certain fraction (~4%) of familial ALS[5]. Despite its correlation with and causation of ALS, the role of TDP-43 in ALS pathogenesis has been largely unknown at the mechanistic level.

Multimerization of TDP-43 underlies its physiological and pathological roles. Under normal physiological conditions, homo-oligomerization of TDP-43 occurs through its N-terminal domain and is necessary for its RNA regulatory functions, such as splicing[6–8]. At the C-terminus TDP-43 contains an intrinsically disordered region (IDR) with prion-like glutamine/asparagine-rich (Q/N) and glycine-rich regions, which can undergo liquid–liquid phase separation (LLPS) to form dynamic protein droplets[9]. The TDP-43 IDR mutations that are linked to familial ALS cases enhance intrinsic aggregation propensity and protein stability of TDP-43[10,11] and result in altered phase separation[9], which could contribute to disease propagation through acceleration of the formation and accumulation of pathological aggregates[12–14]. The modular architecture of TDP-43 has led to several hypotheses that its N-terminus-dependent oligomerization modulates C-terminal IDR-mediated aggregation either by enhancing[15] or hindering IDR interactions between adjacent TDP-43 molecules[6,16].

The severity of TDP-43 toxicity is correlated with the levels of wild-type and mutant TDP-43 expression in the various cellular and animal models[17–26]. However, cytoplasmic TDP-43 aggregation is not always detectable in these models. Moreover, in a certain type of degenerating upper motor neurons, loss of nuclear TDP-43 can occur without the accumulation of cytoplasmic aggregates[27]. Therefore, it has been difficult to evaluate how TDP-43 aggregation contributes to TDP-43 toxicity. Under these circumstances, it is necessary to develop a system to induce TDP-43 aggregation conditionally. Recently, light-dependent aggregation of *Arabidopsis* cryptochrome-2 was applied to the formation of IDR droplets via LLPS in a light illumination-dependent manner[28]. This optogenetic approach has been successfully extended to the induction of cytotoxic TDP-43 aggregates formation in cultured cells[29,30]. However, interconversion of normal and toxic TDP-43 forms with spatiotemporal precision has not been achieved in animal models yet, which is central for the understanding of TDP-43 toxicity in vivo.

In the present study, we develop an optogenetic TDP-43 variant (opTDP-43) carrying a light-dependent oligomerization module of cryptochrome-2 attached to the IDR, and analyze the mechanisms of TDP-43 toxicity in spinal motor neurons in vivo. Transgenic expression and light stimulation of opTDP-43 in transparent zebrafish larvae show that oligomerization and aggregation of opTDP-43 is inducible and tunable in vivo by external light illumination. We reveal that, in the spinal motor neurons, short-term light illumination reversibly increases the cytoplasmic opTDP-43 pool and elevates myofiber denervation frequency in the absence of distinct aggregate formation. Furthermore, longer chronic light stimulation eventually leads to accumulation of cytoplasmic opTDP-43 aggregates that further seed aggregation of non-optogenetic TDP-43, which is accompanied by motor decline. The sequential pathological alterations of spinal motor neurons triggered by opTDP-43 oligomerization may provide clues about how motor neuron degeneration progresses at both molecular and cellular levels in a prodromal phase of ALS.

## Results

**Toxicity caused by TDP-43 overexpression**. To explore mechanisms of TDP-43 toxicity associated with its cytoplasmic aggregation in spinal motor neurons, we first aimed to induce TDP-43 aggregation by its overexpression in the caudal primary motor neurons (CaPs) of zebrafish, which innervate a ventral third of the myotome and are present uniquely in every spinal hemisegment (Fig. 1a, b)[31]. We generated a Gal4-inducible transgene of the zebrafish *tardbp*, encoding one of the two zebrafish TDP-43 paralogues, tagged with mRFP1 at its N-terminus (mRFP1-TDP-43z) (Fig. 1c). To test the functionality of mRFP1-TDP-43z as TDP-43, we generated knock-out alleles for both *tardbp* and its paralogue *tardbpl* with the CRISPR-Cas9 system (i.e. *tardbp-n115* and *tardbpl-n94*, respectively, Supplementary Fig. 1a–c). The TDP-43 double knock-out (DKO) embryos exhibited a blood circulation defect at 24–48 h post-fertilization (hpf) (Supplementary Fig. 1d) and were lethal[32]. We injected mRNAs encoding wild-type Tardbp and mRFP1-TDP-43z into the TDP-43 DKO embryos at the one-cell stage and found that the blood circulation defect was rescued by both (Supplementary Fig. 1e, Supplementary Movie 1), indicating that mRFP1-TDP-43 is functional. We then overexpressed mRFP1-TDP-43z in CaPs by combining Tg[UAS:mRFP1-TDP-43z] with the Tg[SAIG213A] driver (Fig. 1a, b)[33], and analyzed their muscle innervation. The mRFP-TDP-43z overexpression significantly reduced the total axonal length at 48 hpf (Fig. 1d–f, Supplementary Movie 2), while the axon arborized within the inherent innervation territory of the ventral myotomes (Fig. 1d, e) and their branching frequency (i.e. branching as calculated per total axon length) were comparable to that of the wild-type CaP (Fig. 1g), showing that overexpression of mRFP-TDP-43z primarily affects axon outgrowth, but not pathfinding or branching. However, the overexpressed mRFP-TDP-43z was predominantly accumulated in the nucleus and cytoplasmic aggregation was undetectable in the CaP at 48 hpf (Fig. 1h). These observations suggest that an elevated level of TDP-43 causes neurotoxicity independently of cytoplasmic aggregation in the spinal motor neurons.

**A photo-switchable TDP-43: opTDP-43**. Next, we developed an alternative strategy to induce cytoplasmic TDP-43 aggregation. The IDR at the C-terminal of TDP-43 has a high propensity to form aggregates[10]. Therefore, we reasoned that cytoplasmic TDP-43 aggregation might effectively occur when the proximity between IDRs was increased by addition of an exogenous multimerization tag, such as *Arabidopsis* cryptochrome CRY2. We first assessed the oligomerization capacity of mRFP1-tagged CRY2olig in the skeletal muscle cells by taking advantage of their relatively large nucleus and cytoplasm. We created a transgenic zebrafish line carrying a fusion of mRFP1 and CRY2olig, a point mutant version of CRY2 (E490G) that exhibits significant clustering upon blue light illumination[34], under the UAS sequence (Fig. 2a; Tg[UAS:mRFP1-CRY2olig]). When driven by the ubiquitous Gal4 driver Tg[SAGFF73A], mRFP1-CRY2olig was dispersed throughout the elongated skeletal muscle cells under dark conditions at 30 hpf (Fig. 2b). Upon blue light illumination via confocal laser scanning, mRFP1-CRY2olig instantaneously formed clusters with varying sizes throughout the cells within 30 min, showing that CRY2olig clustering is efficiently inducible by external light illumination. Then, to adopt the clustering capacity of mRFP1-CRY2olig to TDP-43, we inserted the zebrafish *tardbp* between the mRFP1 and CRY2olig modules, and

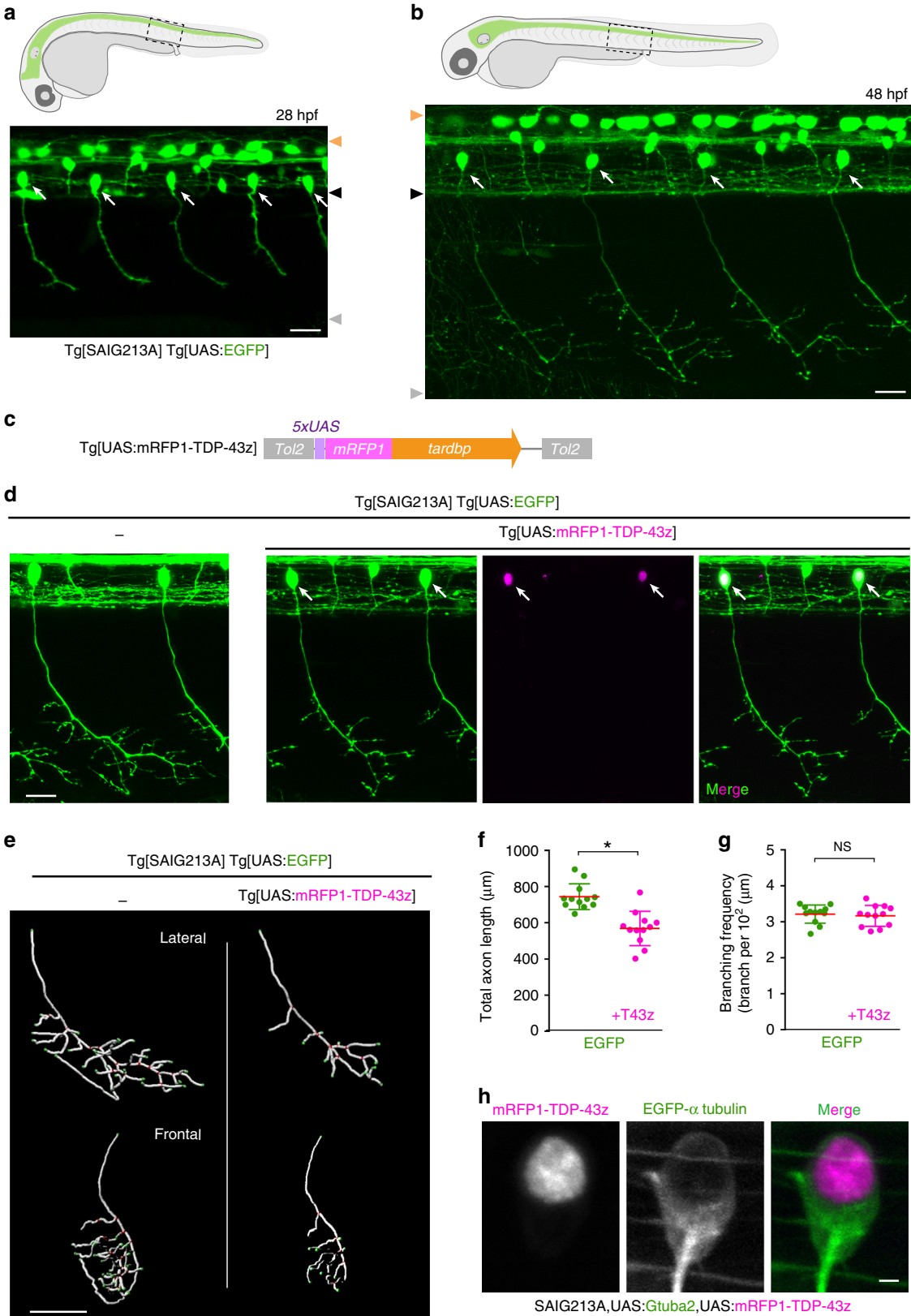

designated the resulting mRFP1-*tardbp*-CRY2olig fusion gene as opTDP-43z (i.e., optogenetic TDP-43 of zebrafish) (Fig. 2a). When expressed via mRNA injection at the one-cell stage, opTDP-43z rescued the blood circulation defect of TDP-43DKO embryos under dark conditions as efficiently as wild-type *tardbp* (Supplementary Fig. 1e, Supplementary Movie 1), confirming that

opTDP-43z is functional. Since the strong whole-body expression of mRFP-TDP-43z driven by Tg[SAGFF73A] driver perturbed development (Supplementary Fig. 2), we generated a UAS transgenic line that expressed a tolerable level of opTDP-43z in combination with Tg[SAGFF73A] driver (Fig. 2a; Tg[UAS: opTDP-43z]). In Tg[SAGFF73A] Tg[UAS:opTDP-43z] double

**Fig. 1 TDP-43 overexpression halts axon outgrowth independently of cytoplasmic aggregation. a, b** CaPs (arrows) in Tg[SAIG213A] Tg[UAS:EGFP] fish. Orange, black and gray arrowheads indicate dorsal and ventral limits of the spinal cord, and ventral myotomal borders, respectively. **c** The structure of Tg[UAS:mRFP1-TDP-43z]. **d** CaPs in Tg[SAIG213A] Tg[UAS:EGFP] (left) and Tg[SAIG213A] Tg[UAS:EGFP] Tg[UAS:mRFP1-TDP-43z] (right) larvae at 48 hpf. Arrows indicate CaPs expressing mRFP1-TDP-43z. **e** The lateral and frontal views of skeletonized CaP axons of Tg[SAIG213A] Tg[UAS:EGFP] (left) and Tg[SAIG213A] Tg[UAS:EGFP] Tg[UAS:mRFP1-TDP-43z] (right). The axon branch points and terminals are indicated by red and green, respectively. (See also Supplementary Movie 1). **f, g** Total length and branching frequency of CaP axons at the spinal segment 14–17 of Tg[SAIG213A] Tg[UAS:EGFP] (green, 12 cells, 5 animals) and Tg[SAIG213A] Tg[UAS:EGFP] Tg[UAS:mRFP1-TDP-43z] (magenta, 12 cells, 6 animals). *$p < 0.0001$ (unpaired $t$-test, two-tailed). NS, not statistically significant ($p = 0.66$). **h** Localization of mRFP1-TDP-43z in a CaP of Tg[SAIG213A] Tg[UAS:Gtuba2] Tg[UAS:mRFP1-TDP-43z] at 48 hpf. EGFP-tagged α-tubulin expressed from Tg[UAS:Gtuba2][64] serves as a marker for the cytoplasm. The bars indicate 20 μm (**a**, **b**, **d**), 40 μm (**e**), and 2 μm (**h**). Error bars show standard deviation (SD).

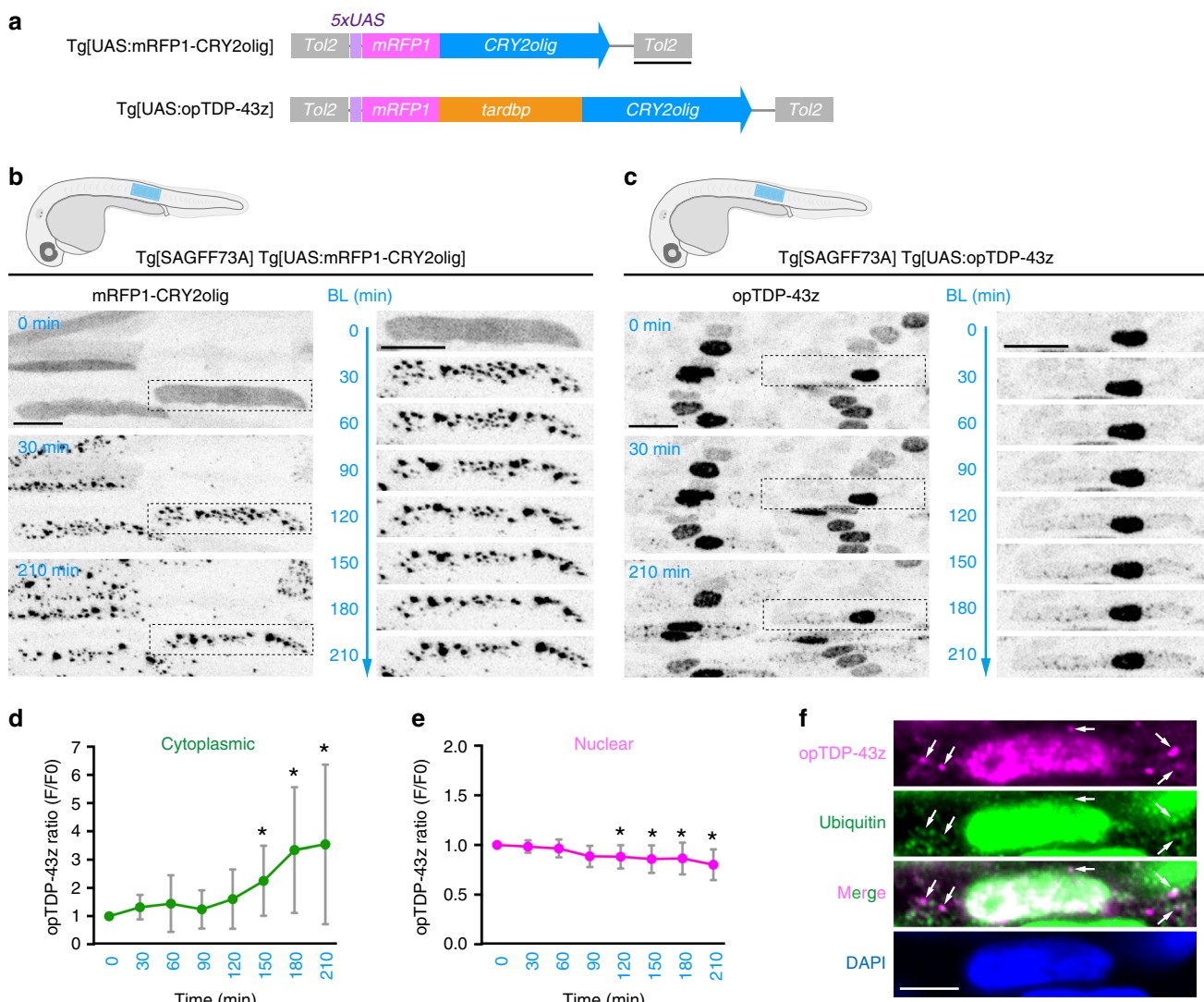

**Fig. 2 A photo-switchable TDP-43: opTDP-43. a** The structures of Tg[UAS:mRFP1-CRY2olig] and Tg[UAS:opTDP-43z]. **b, c** Skeletal muscle of Tg [SAGFF73A] Tg[UAS:mRFP1-CRY2olig] and Tg[SAGFF73A] Tg[UAS:opTDP-43] fish from 28 hpf (0 min) to 31.5 hpf (210 hpf). The blue light was illuminated from 0 to 210 min. Montages of single skeletal muscle cells expressing mRFP1-CRY2olig and opTDP-43z (dashed boxes) are shown on the right. **d, e** The averaged change of opTDP-43z intensity in the cytoplasm (**d**) and nucleus (**e**) of skeletal muscle cells in Tg[SAGFF73A] Tg[UAS:opTDP-43z] fish during the illumination ($N = 8$ cells from the same animal). Error bars show SD, and the center of the error bars is the mean. The asterisks indicate the statistically significant change in opTDP-43z fluorescence intensity at each time point in comparison to the opTDP-43z level at $t = 0$, and adjustments were not made for multiple comparisons. In **d** $p = 0.0125$, 0.010, 0.0234 at 150, 180, 210 min, respectively (unpaired $t$-test, two-tailed). In **e** $p = 0.0092$, 0.0128, 0.0114, 0.0311 at 120, 150, 180, 210 min, respectively. **f** Immunofluorescence of the skeletal muscle of fish illuminated for 3.5 h, using anti-RFP (for opTDP-43z) and anti-ubiquitin antibodies. Arrows indicate the representative of opTDP-43z foci that are partially ubiquitinated. The bars indicate 500 bp (**a**), 20 μm (**b**, **c**), and 5 μm (**f**).

transgenic fish, opTDP-43z predominantly localized to the nucleus of the skeletal muscle cells under dark conditions (Fig. 2c), unlike the widespread localization of mRFP1-CRY2olig, suggesting that opTDP-43z localization is regulated by TDP-43-dependent mechanisms. We found that, while the nuclear-enriched opTDP-43z localization persisted during the 3.5 h of blue light illumination (28–31.5 hpf), the cytoplasmic opTDP-43z gradually increased (Fig. 2c, d, Supplementary Movie 3) and opTDP-43z foci appeared 60–90 min after the initiation of illumination (Fig. 2c). On the other hand, the nuclear opTDP-43z signal decreased slightly but significantly over time during the illumination (Fig. 2e). We also found that the cytoplasmic opTDP-43z foci were partially ubiquitinated as shown by immunofluorescence (Fig. 2f), suggesting that the opTDP-43z level is regulated by proteolysis[35,36]. Altogether, these observations demonstrate that opTDP-43z is a photo-switchable variant of TDP-43 that clusters in a blue light illumination-dependent manner.

**Cytoplasmic opTDP-43z mislocalization by light illumination**. Next, we tested the feasibility of CRY2olig oligomerization in spinal motor neurons in vivo. When driven by Tg[SAIG213A] driver, mRFP1-CRY2olig was dispersed throughout the CaPs at 30 hpf under dark conditions (Supplementary Fig. 3). Upon blue light illumination, mRFP1-CRY2olig rapidly clustered in somas and axons during the first 10 min of illumination, while the mRFP1-CRY2olig clusters gradually disappeared and a homogeneous distribution of mRFP1-CRY2olig was restored, once illumination ceased (Supplementary Fig. 3, Supplementary Movie 4), showing that CRY2olig clustering is rapidly and reversibly controllable by light in the spinal motor neurons in vivo.

Then, we expressed opTDP-43z in the spinal motor neurons, as well as tactile sensing Rohon-Beard (RB) cells, by combining both Tg[mnr2b-hs:Gal4][37] and Tg[SAIG213A] drivers. Under dark conditions, opTDP-43z primarily localized to the nucleus of both cell types at 28 hpf in Tg[mnr2b-hs:Gal4] Tg[SAIG213A] Tg[UAS:opTDP-43z] Tg[UAS:EGFP] quadruple transgenic fish (Fig. 3a). Upon blue light illumination of the spinal cord, the nuclear-enriched localization of opTDP-43z persisted for about 90 min and was gradually expanded to the entire EGFP-positive area in both spinal motor neurons and RB cells (Fig. 3b, c, Supplementary Movie 5), in contrast to the rapid clustering of mRFP1-CRY2olig (Fig. 3d). The gradual expansion of opTDP-43z signal represented its cytoplasmic mislocalization as the signal spread outside the nucleus visualized by EGFP-tagged histone H2A variant (h2afva) (Fig. 3e). Unexpectedly, however, unlike opTDP-43z in the skeletal muscle cells (Fig. 2c), the cytoplasmic mislocalization did not lead to distinct foci formation within the time frames examined (up to 4.5 h illumination), suggesting that the spinal motor neurons and RB cells have lower propensity to form cytoplasmic opTDP-43 foci than the skeletal muscle cells. The cell type-dependent variation of opTDP-43 mislocalization and aggregation was also substantiated by the observations that neither embryonic epithelial cells nor differentiated skeletal muscle fibers displayed cytoplasmic mislocalization or aggregation of opTDP-43z under the same light illumination condition (Supplementary Fig. 4).

**opTDP-43z oligomerization perturbs axon outgrowth**. To explore the impact of light-induced cytoplasmic opTDP-43z mislocalization at the whole cell level, we restricted opTDP-43z expression to CaPs by using Tg[SAIG213A] driver. We devised a protocol by which Tg[SAIG213A] Tg[UAS:EGFP] Tg[UAS:opTDP-43z] triple transgenic fish were raised under continuous

dark conditions until 48 hpf except being illuminated for 3 h during 28–31 hpf (Fig. 4a). Under this paradigm, opTDP-43z was primarily localized within the nucleus at 28 hpf, then dispersed throughout the nucleus and cytoplasm upon illumination, and restored its nuclear-enriched localization at 48–50 hpf (Fig. 4a, b). We found by morphological analyses that total axon length, but not branching frequency, of CaPs decreased at 48–50 hpf by 13% in the fish treated with 3 h of blue light illumination, while such a phenotype was detected neither under continuous dark conditions nor by mRFP1-CRY2olig expression (Fig. 4c–e). As observed with mRFP1-TDP-43z overexpression (Fig. 1), the axons of light-stimulated CaPs expressing opTDP-43z arborized within their inherent ventral innervation territory (Fig. 4c), and their branching frequency remained unchanged (Fig. 4e), suggesting that the light-dependent opTDP-43z toxicity primarily influences axon outgrowth, but not pathfinding or branching.

The absence of distinct cytoplasmic foci formation in the CaPs raises the possibility that opTDP-43z exerts its toxicity through dragging of non-optogenetic TDP-43 out of the nucleus to the cytoplasm. To test this possibility, we constructed Tg[mnr2b-hs:EGFP-TDP-43z] expressing EGFP-TDP-43z under the control of the mnr2b promoter, and established a Tg[SAIGFF213A] Tg[UAS:opTDP-43z] Tg[mnr2b-hs:EGFP-TDP-43z] triple transgenic fish. Under dark conditions at 28 hpf, both opTDP-43z and EGFP-TDP-43z exhibited nuclear localization with subnuclear distribution patterns similar to each other (Fig. 4f). Contrary to our prediction, however, the nuclear-enriched EGFP-TDP-43z localization remained unaffected while opTDP-43z was dispersed throughout the soma with 4 h-blue light illumination during 28–32 hpf (Fig. 4f–i), demonstrating that light-induced opTDP-43z mislocalization occurs independently of the non-optogenetic TDP-43 pool. These observations suggest that the perturbation of axon outgrowth by light-stimulated opTDP-43z is unlikely to be caused by loss of TDP-43 function due to nuclear TDP-43 reduction or depletion.

**Light-stimulated opTDP-43z promotes myofiber denervation**. The opTDP-43z-mediated axon outgrowth defects raised the question as to whether opTDP-43z perturbs axon extension or promotes axon shrinkage, or both. To address this, we analyzed a major axon collateral of CaP innervating the dorsal side of its innervation territory that had experienced tertial branching at 56 hpf (provisionally named dorsal axon collateral of CaP with tertial branching: DCCT) (Fig. 5a). We shifted timing of the 3-h blue light illumination to a later stages (56–59 hpf) and examined whether arborized DCCTs shrinks or not by light stimulation of opTDP-43z at 72 hpf (Fig. 5b). Live imaging revealed that the total DCCT length increased by 26% from 56 to 72 hpf, (Fig. 5c, d) and formed one additional branch on average in Tg[SAIG213A] Tg[UAS:GFP] larvae. Intriguingly, we noticed that a minor but significant population of single CaPs (24%) increased their total DCCT length with a decrease in the number of collateral branches (Fig. 5d, e), indicating that normal DCCT outgrowth involves both extension and shrinkage, as the extension occurs more frequently. The expression of opTDP-43z itself did not affect the average DCCT outgrowth rate and branch number under dark conditions at 56 hpf (Fig. 5d, e). On the other hand, the average DCCT growth rate significantly declined by 11% with at 72 hpf, when CaPs expressing opTDP-43z had been illuminated for 3 h (from 56 to 59 hpf) (Fig. 5d). Remarkably, 28% (5 out of 18 DCCTs) of the illuminated CaPs decreased total DCCT lengths (Fig. 5c), and 44% showed reduced DCCT branch number (Fig. 5e), demonstrating that axon shrinkage contributes to the observed axon outgrowth defects.

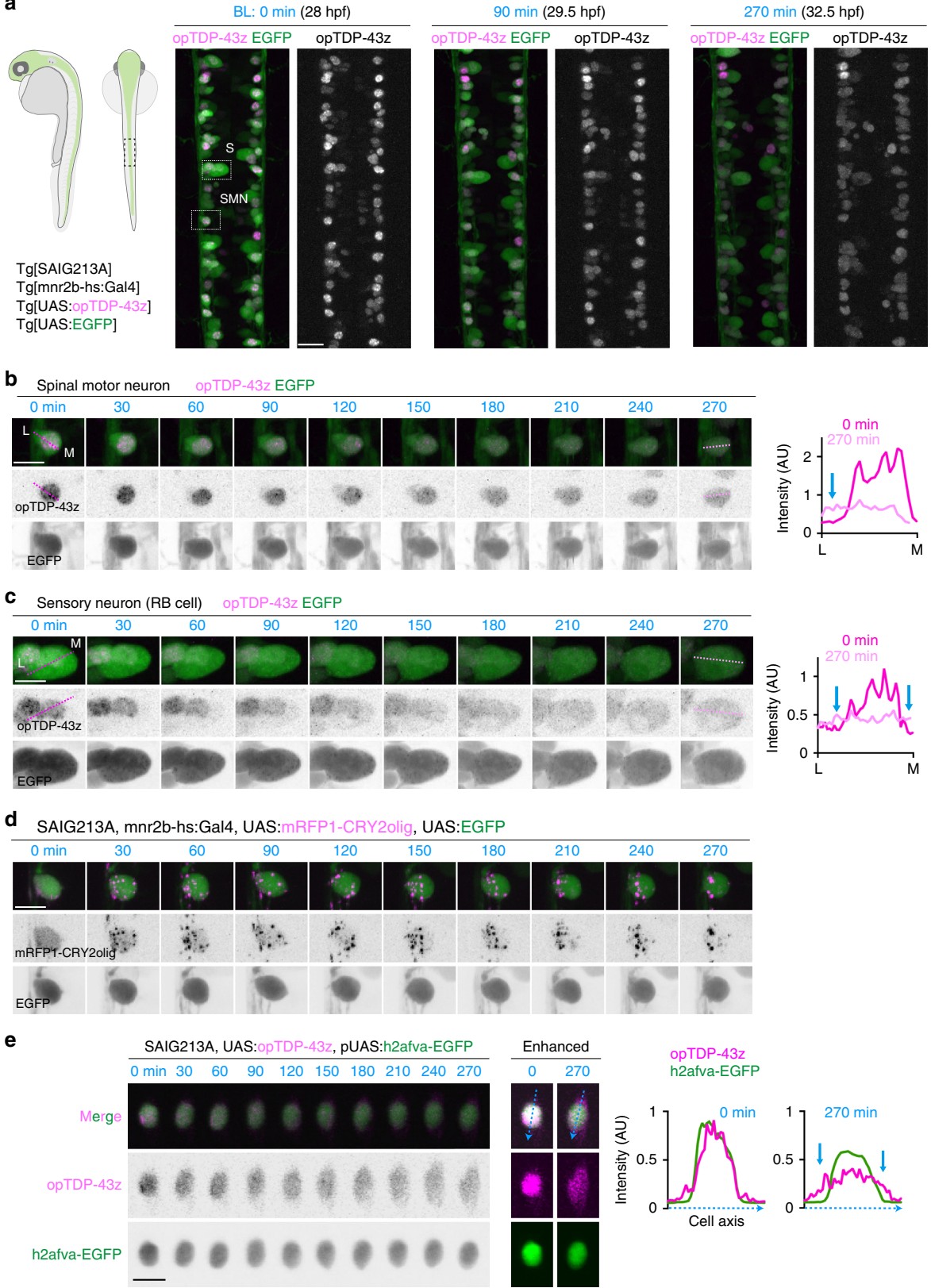

We then investigated whether the DCCT shrinkage involves myofiber denervation by live monitoring of pre- and postsynaptic structures with Vamp2-Venus and tdTomato-tagged acetylcoline receptor (dT-chrnd), respectively (Fig. 5f)[37]. We found that, in both wild-type and opTDP-43z expression conditions prior to light stimulation, the DCCT axon terminals were decorated by Vamp2-Venus, and the Vamp2-Venus signals were well colocalized with dT-chrnd (Fig. 5f–h), indicating normal neuromuscular assembly. Live imaging revealed that, in the opTDP-43z-expressing CaPs after the 3 h of illumination (during 56–59 hpf), the number of the DCCT axon terminals with Vamp2-Venus/dT-chrnd juxtaposition reduced (Fig. 5i, j), demonstrating that the

**Fig. 3 Light illumination-dependent cytoplasmic mislocalization of opTDP-43 in neuronal cells. a** The dorsal view of the spinal cord at the segment 14–17 levels of a Tg[SAIG213A] Tg[mnr2b-hs:Gal4] Tg[UAS:opTDP-43z] Tg[UAS:EGFP] quadruple transgenic fish. A spinal motor neurons (SMN) and a Rohon-Beard sensory neuron (RB cell, S) were highlighted with dashed boxes and analyzed in detail in (**b, c**). **b, c** Montages of the spinal motor neurons and the RB cell during the light illumination. The graphs show the fluorescent intensities of opTDP-43z along the dotted line drawn from the lateral (L) to medial edges (M) of the EGFP signal. The blue arrows indicate the cytoplasmic increase of opTDP-43z. The unit for y-axes are the same between 0 and 270 min. **d** Montage of the spinal motor neuron expressing mRFP1-CRY2olig in the same illumination condition as in (**a**). **e** Cytoplasmic mislocalization of opTDP-43z in Tg[SAIG213A] Tg[UAS:opTDP-43z] fish injected with a plasmid harboring UAS regulated EGFP-tagged histone H2A variant H2afva. The graphs show the fluorescent intensities of opTDP-43z and EGFP-H2afva along blue dotted line drawn across the cell axes. The blue arrows indicate the cytoplasmic increase of opTDP-43z. The bars indicate 20 μm (**a**), 10 μm (**b–e**).

opTDP-43-dependent decrement in DCCT terminal number is accompanied by myofiber denervation. The disappearance of the terminals with Vamp2-Venus/dT-chrnd juxtaposition was also observed in wild-type DCCTs at a lower frequency (Fig. 5i, j, left). Overall, these observations show that the DCCT shrinkage is associated with myofiber denervation, and that optogenetic TDP-43 oligomerization raises the denervation frequency.

**Longer illumination causes cytoplasmic opTDP-43 aggregation.** Targeted optogenetic stimulation via confocal laser scanning required the fish to be agarose-embedded, which restricted the illumination duration to a maximum of ~4 h to fully maintain fish viability. Therefore, we tested whether a longer-term illumination promoted formation of opTDP-43 aggregates in the cytoplasm of spinal motor neurons. For that purpose, we constructed transgenic fish in which most of the spinal motor neurons expressed a CRY2olig-tagged human TDP-43 (opTDP-43h, for optogenetic TDP-43 of human) from an *mnr2b*-BAC transgene (Tg[mnr2b-hs:opTDP-43h]) (Fig. 6a) and established a system for longitudinal field illumination of blue LED light against unrestrained fish. The Tg[mnr2b-hs:opTDP-43h] and Tg[mnr2b-hs:EGFP-TDP-43z] transgenes were combined to allow for simultaneous live monitoring of opTDP-43h and non-optogenetic TDP-43 in the spinal motor neurons. Prior to the illumination at 2 days post-fertilization (dpf), both opTDP-43h and EGFP-TDP-43z were primarily localized in the nucleus (Fig. 6b). We found that, 24 h after the illumination, opTDP-43h was dispersed throughout the cell and formed foci in the cytoplasm at 72 hpf. The foci formation was further enhanced over the subsequent 48 h of illumination (i.e., 72–120 hpf). Despite the distinct cytoplasmic opTDP-43h mislocalization and foci formation, EGFP-TDP-43z was predominantly localized to the nucleus at 72 hpf, suggesting that opTDP-43h mislocalization and foci formation was initiated independently of EGFP-TDP-43z. Intriguingly, at later time points (e.g., 96–120 hpf), focal EGFP-TDP-43z signals colocalized with large opTDP-43h foci (Fig. 6c). The prolonged blue light illumination alone neither induced cytoplasmic mislocalization nor foci formation of EGFP-TDP-43z, in the absence of opTDP-43h (Supplementary Fig. 5). These observations raise a possibility that a long-term illumination turns opTDP-43h into aggregates that incorporate non-optogenetic TDP-43. Then, to probe molecular dynamics of opTDP-43h during blue light illumination, we performed fluorescence recovery after photobleaching (FRAP) experiments. Under continuous dark conditions, opTDP-43h predominantly localized in the nucleus at 5 dpf (Fig. 6d left). Upon bleaching a fraction of the nuclear opTDP-43h signal, we observed a strong recovery of the signal within the following 30 min (Fig. 6e), showing that the nuclear opTDP-43h is highly mobile. In contrast, cytoplasmic opTDP-43h foci that had been induced by 72-h blue light illumination and incorporated EGFP-TDP-43z rarely displayed fluorescence recovery after bleaching (Fig. 6d right, e), indicating slow molecular exchange of opTDP-43h. Taken together, these observations suggest that the light-induced

cytoplasmic opTDP-43h foci represent immobile protein assemblies (aggregates) with self-seeding capacity of TDP-43 aggregation.

**An IDR mutation enhances protein stability and toxicity.** The gradual mislocalization and aggregation of non-optogenetic TDP-43 promoted by the light-stimulated opTDP-43h prompted us to hypothesize that opTDP-43h first oligomerizes via the CRY2olig module and subsequently seeds non-optogenetic TDP-43 aggregation via its aggregate-prone IDR[10,38]. To prove contribution of the IDR to oligomerization-dependent toxicity of opTDP-43h, we created an opTDP-43 mutant with an IDR mutation (A315T) linked to familial ALS (opTDP-43h^A315T), occurring in the residue implicated in irreversible aggregation[13], and expressed opTDP-43h^A315T widely in the spinal motor neurons from Tg[mnr2b-hs:opTDP-43h^A315T] (Fig. 7a, b). In Tg[mnr2b-hs:opTDP-43h^A315T] Tg[mnr2b-hs:EGFP-TDP-43z] double transgenic fish, opTDP-43h^A315T displayed nuclear-enriched localization prior to the LED illumination at 2 dpf (Fig. 7b). Within 24 h of illumination, opTDP-43h^A315T mislocalized to the cytoplasm and formed aggregates. Seeding of non-optogenetic EGFP-TDP-43z occurred in virtually all of large cytoplasmic opTDP-43h^A315T aggregates, as was the case with opTDP-43h (Fig. 7c). It should be noted that the expression level of opTDP-43h^A315T protein was less than that of opTDP-43h in the Tg[mnr2b-hs:opTDP-43h] fish (Fig. 7a, b), at least partly due to the lower level of mRNA (Fig. 7d). The illuminated fish expressing opTDP-43h were viable with seemingly normal free-swimming activity at 5–6 dpf, suggesting that the toxicity associated with opTDP-43h has only a minor effect at the behavioral level. On the other hand, 13% of the illuminated fish expressing opTDP-43h^A315T, but none of the non-illuminated siblings, failed to inflate the swim bladder at 5 dpf and showed declined locomotor ability (Fig. 7e, Supplementary Movie 6), indicating that the IDR contributes to the oligomerization-dependent opTDP-43h toxicity.

To explore mechanisms underlying the A315T mutation-dependent toxicity, we first compared molecular compositions of opTDP-43h and opTDP-43h^A315T aggregates. We detected weak, but distinct, immunoreactivity for phosphorylated TDP-43 at serines 409/410 in about ~60% of spinal motor neurons containing opTDP-43h or opTDP-43h^A315T aggregates, indicating the occurrence of pathological C-terminus phosphorylation of TDP-43 at a similar level in both conditions (Fig. 7f). We also tested whether components of conventional stress granules (SGs) were recruited to these aggregates by using the antibodies against human G3BP and TIAL1 that robustly co-labeled heat shock-induced SGs in zebrafish (Supplementary Fig. 6)[39,40]. The immunoreactivity for anti-G3BP antibody was detected in about 65% of the cells containing large aggregates of opTDP-43h or opTDP-43h^A315T (Fig. 7g), while that for anti-TIAL1 antibody was observed weakly and much less frequently (~20%, Fig. 7h, i), suggesting that the opTDP-43h aggregates are heterogeneous protein assemblies and that the A315T mutation might have a minor impact on their protein compositions. Moreover, given the

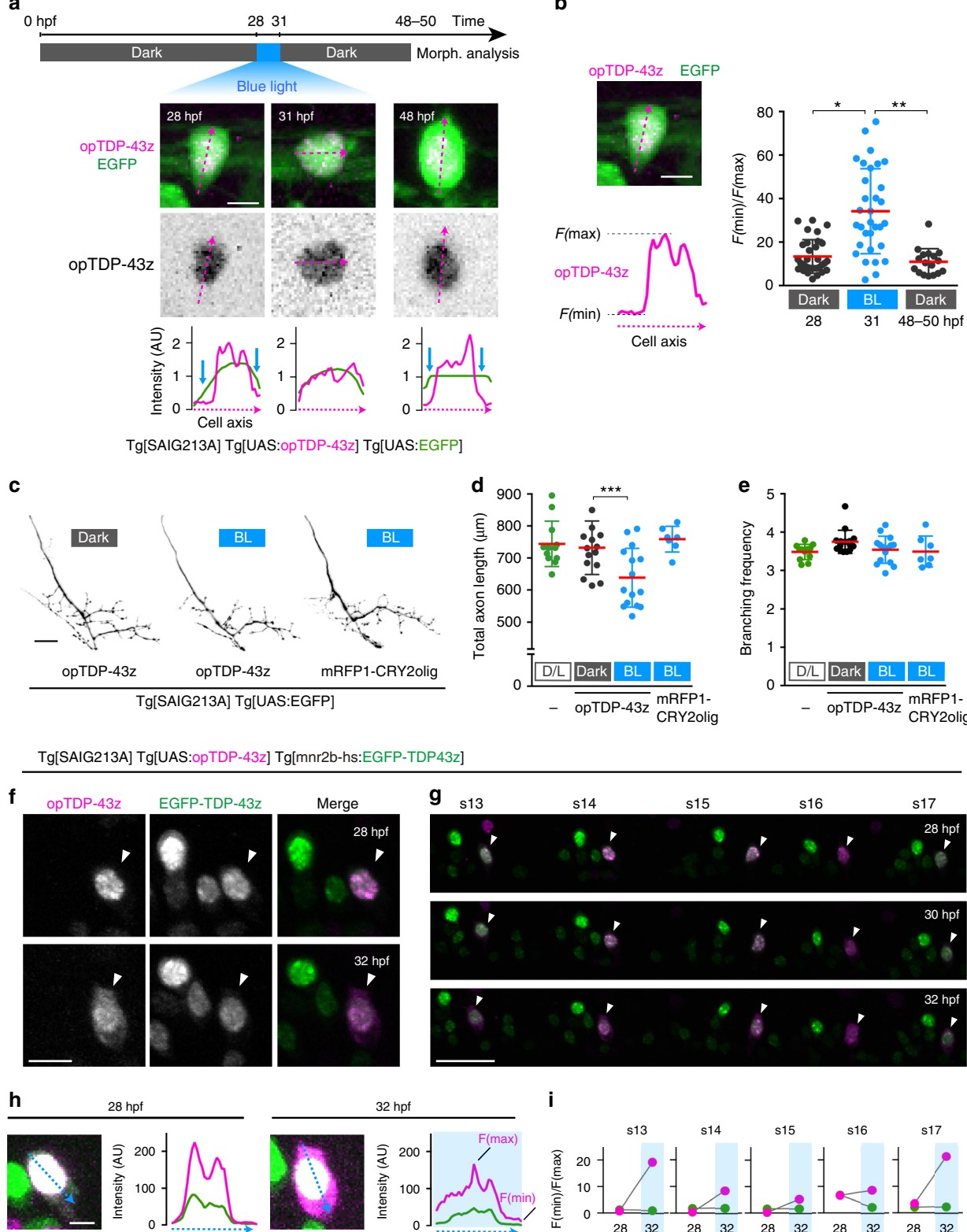

consistent recruitment of EGFP-TDP-43z to opTDP-43h aggregates (Fig. 7i), these results suggest that neither phosphorylation at S409/S410 nor SGs is essential for self-seeding of TDP-43 aggregation. We then pursued a possibility that the A315T mutation affected the stability of TDP-43 protein in the spinal motor neurons. We measured the intensity of opTDP-43h signal in the somas of single spinal neurons before and after 24-hour

illumination, during which cytoplasmic opTDP-43h mislocalization and aggregation occurred, by using EGFP-TDP-43z signal as an internal control (Fig. 7a, b). The relative intensity of opTDP-43h to EGFP-TDP-43z remained largely unchanged before and after the illumination (Fig. 7J). Importantly, despite the lower level of opTDP-43h[A315T] expression (Fig. 7a, b), the relative intensity of opTDP-43h[A315T] to EGFP-TDP-43z was significantly

**Fig. 4 Cytoplasmic opTDP-43z mislocalization is accompanied by diminished axon outgrowth. a** The light-illumination paradigm of CaPs. The spinal cord of Tg[SAIG213A] Tg[UAS:opTDP-43z] Tg[UAS:EGFP] fish at the segment 13–18 level were illuminated with a blue laser, and CaPs were subjected to morphological analysis at 48–50 hpf. A single CaP was analyzed from dorsal (28, 31 hpf) and lateral (48 hpf) views. The fluorescence intensity along the longest inner diameter (dashed magenta arrow) is plotted at each time point. Blue arrows indicate the presumptive cytoplasmic area, where the opTDP-43z signal is faint. **b** Cytoplasmic shift of opTDP-43 is evaluated as a relative value of minimal ($F$(min), cytoplasm) and maximal ($F$(max), nuclear) fluorescence intensity along the longest inner diameter (dashed magenta line). The results were obtained from 32 cells (28, 31 hpf) and 17 cells (48 hpf) in three independent fish. *$p < 0.0001$, **$p < 0.0001$ (unpaired $t$-test, two-tailed). **c** Axons of CaPs expressing opTDP-43z with (BL, middle) or without (Dark, left) blue light stimulation and mRFP1-CRY2olig with blue light stimulation (right). **d, e** The total axon length and branching frequency of CaP axons in Tg[SAIG213A] Tg[UAS:EGFP] fish raised under normal laboratory light-dark cycle (L/D, the same data sets in Fig. 1e, f), Tg[SAIG213A] Tg[UAS:opTDP-43z] Tg[UAS:EGFP] fish with (BL, 15 cells, 4 animals) or without (Dark, 15 cells, 4 animals) blue light stimulation, and Tg[SAIG213A] Tg[UAS:mRFP1-CRY2olig] Tg[UAS:EGFP] fish with the stimulation (7 cells, 2 animals). ***$p = 0.0068$. **f, g** CaPs (arrowhead) and other *mnr2b*-positive motor neurons in the segment 14 (**f**) and 13–17 (**g**) of Tg[SAIG213A] Tg[UAS:opTDP-43z] Tg[mnr2b-hs:EGFP-TDP43z] fish that was illuminated with a blue light during 28–32 hpf. **h, i** Cytoplasmic shift of opTDP-43z and EGFP-TDP-43z in the CaP in (**f**). The fluorescence intensities of opTDP-43z (magenta) and EGFP-TDP-43z (green) were plotted along the blue dashed arrows (**h**). Images shown are enhanced to identify soma outline. **i** The relative intensity of cytoplasmic signal ($F$(min)/$F$(max)) for opTDP-43z (magenta) and EGFP-TDP-43z (green) in each spinal segment. The bars indicate 5 μm (**a, b**), 10 μm (**f, h**), 20 μm (**c**), and 30 μm (**g**). Error bars show SD.

increased after illumination compared with that of opTDP-43h (Fig. 7j). These observations suggest that the A315T mutation increases stability of TDP-43 protein in the spinal motor neurons, consistent with the previous observations in cultured neurons[11,41], and imply that the A315T mutation causes toxicity by generating stable TDP-43 oligomers through modulating IDR-mediated oligomerization.

## Discussion
Pathological aggregation of TDP-43 via the IDRs is proposed to be antagonized by N-terminal-mediated homo-oligomerization under physiological conditions[6,16]. In this study, we successfully developed CRY2olig-mediated TDP-43 oligomerization system in vivo and demonstrated CRY2olig-mediated oligomerization led to the accumulation of cytoplasmic opTDP-43 aggregates in the zebrafish spinal motor neurons. This CRY2olig-driven opTDP-43 oligomerization would initially generate reversible interactions within the IDRs, some of which occasionally transform into an irreversible form. Then, such irreversible "knots" of opTDP-43 eventually seed IDR-mediated aggregation of non-optogenetic TDP-43 (Fig. 8). Under our illumination conditions, the spinal motor neurons require up to 3 h to fully disperse opTDP-43 throughout the cell, 24 h to accumulate distinct cytoplasmic opTDP-43 aggregates, and several additional days to develop cytoplasmic opTDP-43 aggregates containing non-optogenetic TDP-43. This sequentially regulated illumination-triggered TDP-43 knot and aggregate formation enables direct observation of spinal motor neuron pathology as triggered by IDR-mediated TDP-43 oligomerization. We propose that this opTDP-43-triggered pathology may correspond to a fast-forwarding of spinal motor neuron degeneration in ALS, in which a majority of cases are believed to involve IDR-mediated TDP-43 aggregation, yet currently allows very restricted anatomical access.

We discovered that the reversible cytoplasmic opTDP-43 mislocalization induced by short-term light illumination was sufficient to cause defective motor axon outgrowth accompanied by enhanced myofiber denervation. The physiological nuclear-enriched TDP-43 localization is sustained by nucleocytoplasmic transport system[35,36] as well as protein degradation systems in the cytoplasm, such as the ubiquitin-proteasome system (UPS) and autophagy[42,43]. Since the light-dependent cytoplasmic opTDP-43 mislocalization commences without affecting non-optogenetic TDP-43 localization, the toxicity accompanied by opTDP-43 mislocalization may not be caused by a global shut-down of nucleocytoplasmic transport or proteolysis systems for TDP-43, but rather by dysregulation of RNAs and/or proteins

being bound by opTDP-43. TDP-43 can associate with more than 6,000 RNA targets[44–47], and RNA-binding is antagonistic to toxic TDP-43 oligomerization in an optogenetic cellular model[29], implying that light-dependent opTDP-43 oligomerization would profoundly affect its RNA-binding capacity, thereby influencing the expression of a myriad of genes. In terms of the axon out-growth defect, whether toxicity is ascribable to dysregulation of specific key proteins[42,48] or to a widespread translational abnormality[49], which could lead to stress-inducing misfolded protein accumulation, remains to be investigated. Nevertheless, the normal motor axon pathfinding and unaffected branching frequency suggest a certain specificity of opTDP-43 toxicity and would favor the idea that the cellular growth pathway is primarily affected by the toxicity. It should also be noted that our current illumination paradigm encompasses not only the somas but also the motor axons. Therefore, "resident" cytoplasmic opTDP-43, such as that included in mRNP granules undergoing axonal transport[50] and in mitochondria at the axon terminals[51,52], could also be photoconverted in situ, thereby contributing to the acute toxicity that involves neuromuscular synapse destabilization. Spatially-resolved light stimulation, a major advantage of optogenetics, could identify such potentially multiple pathogenic origins in the future.

The light-dependent cytoplasmic opTDP-43 mislocalization provides unexpected insight into the relationships between TDP-43 multimerization, localization, and toxicity, given that a toxic level of mRFP1-TDP-43 overexpression led neither to cyto-plasmic mislocalization nor aggregation. The persistent nuclear localization of overexpressed mRFP1-TDP-43 indicates a robustness of the nucleocytoplasmic transport and cytoplasmic protein degradation systems against proteostatic perturbation of TDP-43. On the other hand, these TDP-43 surveillance systems appear to be inert against the IDR-mediated TDP-43 oligomers, as evidenced by the light-dependent cytoplasmic opTDP-43 mislocalization. These manipulations of proteostasis and multi-merization revealed that a dosage increase of TDP-43 does not immediately lead to IDR-mediated oligomerization in the spinal motor neurons, and therefore TDP-43 toxicity associated with proteostatic abrogation could be mechanistically distinct from that caused by IDR-mediated oligomerization. Importantly, our animal model approach explicitly revealed a striking cell-type variation of opTDP-43 mislocalization and aggregation, such that the neuronal cells are less prone to accumulate opTDP-43 aggregates compared to the differentiating muscle cells, while both of these cell types are inherently more competent for cyto-plasmic mislocalization than the epithelial cells. Although the mechanisms underlying this cell-type specificity remain

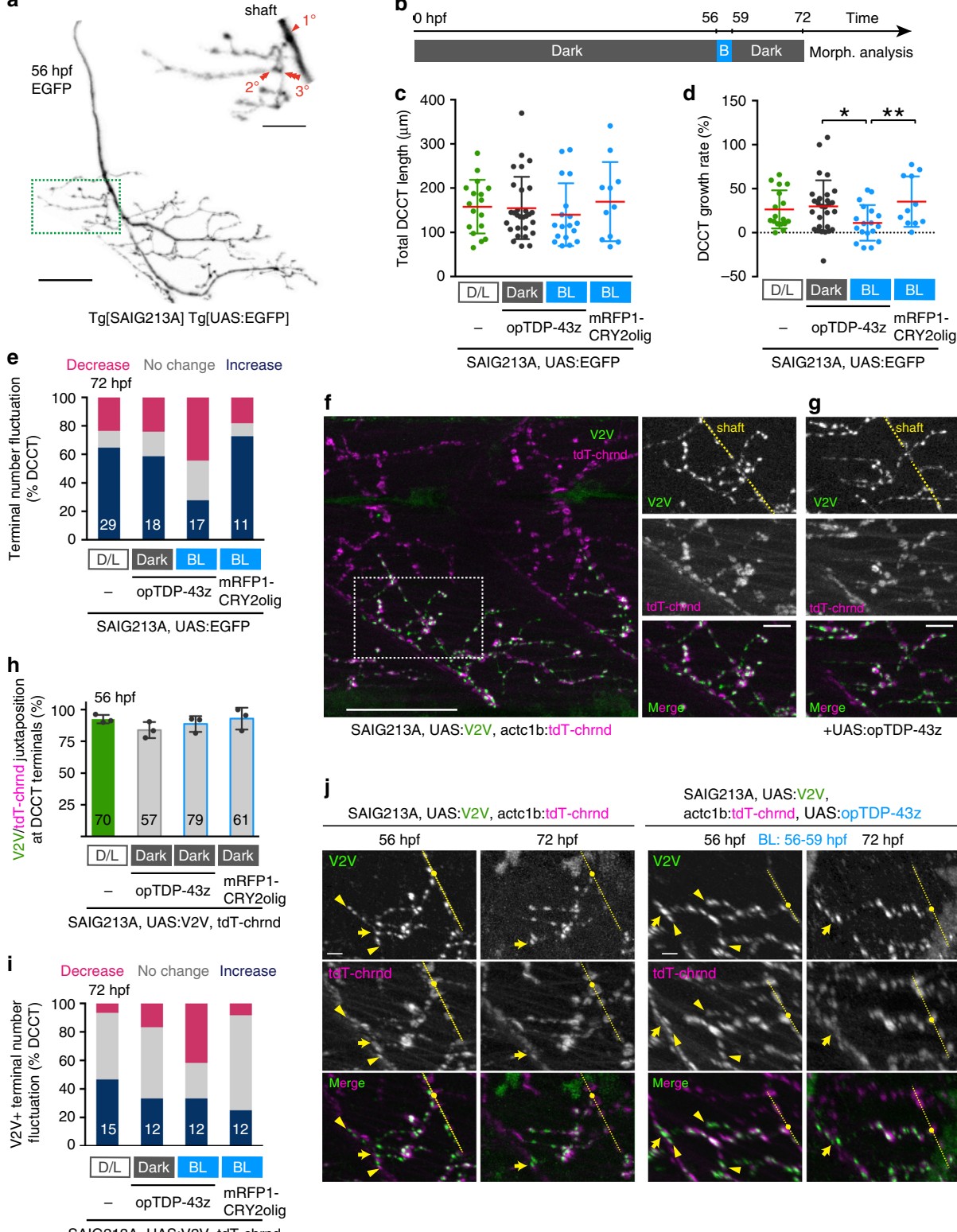

unknown, the present and previous observations[53] emphasize the importance of studying vulnerable cell types in vivo for accurately disclosing the mechanisms underlying TDP-43 localization, and thereby toxicity.

The verification that CRY2olig-mediated opTDP-43 oligomerization is toxic to the spinal motor neurons instead made it difficult to evaluate the toxicity derived from opTDP-43 aggregates alone, as the oligomeric and aggregate forms of opTDP-43

coexists during illumination. In this regard, it is noteworthy that the opTDP-43h^A315T that was expressed less and formed fewer cytoplasmic aggregates was more toxic than the opTDP-43h expressed in larger amounts (Fig. 7), providing an in vivo example in which the amount of accumulated TDP-43 aggregates does not necessarily predict the degree of TDP-43 toxicity[17,18]. It was recently proposed that TDP-43 adopts both reversible and irreversible β-sheet aggregates that are involved in the formation

**Fig. 5 Axonal shrinkage and myofiber denervation caused by opTDP-43z. a** A CaP motor axon of Tg[SAIG213A] Tg[UAS:EGFP] fish. DCCT (green box) was magnified on the right. Primary, secondary and tertial branchings were indicated in red. **b** Light-illumination paradigm. **c–e** Total length (**c**), growth rate (**d**) and fluctuation of axon terminal number (**e**) of DCCTs. Results were obtained from 5 independent animals in opTDP-43z/Dark condition and otherwise from 3 animals. The numbers in the histograms are total numbers of the cells examined. *$p = 0.0224$, **$p = 0.0128$ (unpaired $t$-test, two-tailed). **f** The lateral view of the trunk of Tg[SAIG213A] Tg[UAS:V2V] Tg[actc1b:tdT-chrnd] fish (left) and neuromuscular synapses of the DCCT (right) at 56 hpf. The dashed yellow lines indicate the CaP axon shaft. **g** Neuromuscular synapses of a DCCT in Tg[SAIG213A] Tg[UAS:opTDP-43z] Tg[UAS:V2V] Tg[actc1b:tdT-chrnd] fish at 56 hpf. **h, i** Occurrence of Vamp2-Venus/ tdT-chrnd juxtaposition prior to illumination at 56 hpf (**h**) and fluctuation of terminal number with Vamp2-Venus/tdT-chrnd juxtaposition at 72 hpf (**i**). The numbers in the histograms show the total numbers of axon terminals (**h**) and DCCTs (**i**) that were examined. Results were obtained from 4 independent animals in D/L condition and otherwise from 3 animals. **j** Live imaging of DCCT neuromuscular synapses. Yellow arrowheads indicate the neuromuscular synapses that were not present at 72 hpf. The yellow dashed lines, dots, arrows indicate axon shafts, primary branching points, and contact sites with the myotomal boundaries of CaPs, respectively. Z-stacks are produced from 3D-rotated images made by Imaris, to make the denervation events (arrowhead) clearly visible (**j**, right). The bars indicate 10 μm (**a** top, **f** right, **g**), 25 μm (**a** bottom, **f** left), 5 μm (**j**). Error bars show SD.

of membraneless organelles, such as SGs and pathogenic amyloids, respectively, and that ALS mutations, including A315T, can promote the transition of such reversible to irreversible pathogenic aggregation[13]. Also, in frontotemporal lobar degeneration (FTLD), TDP-43 displays distinct aggregate assemblies and toxic effects in disease-subtype-specific manners[54]. Consistent with these results, we found that cytoplasmic opTDP-43 aggregates were heterogeneous as the phosphorylation at S409/410 and recruitment of G3BP and TIAL proteins occurred with varying degrees. Although TDP-43 toxicity can be independent of phosphorylation at S409/410[29] and recruitment of SG components[55] in cellular models, the toxicity of opTDP-43 aggregates should be evaluated by considering such heterogeneity, and therefore remains as a challenging but important question to be addressed. Alternatively, the seeding capacity for non-optogenetic TDP-43 aggregation suggests that the toxicity of opTDP-43 aggregates can be exerted as a long-term effect through gradual depletion of the available nuclear and cytoplasmic TDP-43 pools into these aggregates, which would manifest as a TDP-43 loss-of-function phenotype[55].

It has been estimated that, even during healthy aging, the spinal motor neurons are substantially lost[56–58]. As a result, the surviving motor units are enlarged to preserve maximal force-generating capacity by compensatory collateral reinnervation[59]. ALS has also been characterized by an elevated number of muscle fibers innervated by a single subterminal axon[60], which is likely a remnant of such compensatory collateral reinnervation events. In the present study, we found by live imaging of axon collateral that an innervation territory of healthy spinal motor neurons is determined by a balance between assembly and disassembly of neuromuscular synapses in zebrafish. We further discovered that optogenetic opTDP-43 oligomerization could tip the balance toward disassembly and decrease the total collateral length. Therefore, our results predict that, once a cellular concentration of IDR-mediated TDP-43 oligomers reaches a critical level, a spinal motor neuron would begin to reduce its motor unit size through repetition of incomplete denervation/reinnervation cycles. Such neurons would also be defective in complementing damaged neighboring motor units through collateral reinnervation, which would accelerate the manifestation of motor decline. We envision that opTDP-43 allows for approaching the mechanisms underlying such dynamic innervation/reinnervation balancing of spinal motor axons in health and TDP-43-associated pathology, as well as for interrogating how not only motor neurons but also diverse types of surrounding cells, including muscle, glial and endothelial cells, respond to and modify TDP-43 toxicity. Moreover, combined with the feasibility of high-throughput, whole organism chemical screening in zebrafish, opTDP-43-mediated motor neuron pathogenesis should be extended for exploring small molecules that restore a normal denervation/

reinnervation balance for spinal motor neurons, which might serve as drugs for ALS and other TDP-43 proteinopathy.

## Methods

**Fish lines.** This study was carried out in accordance with the Guide for the Care and Use of Laboratory Animals of the Institutional Animal Care and Use Committee (IACUC, approval identification number 24-2) of the National Institute of Genetics (NIG, Japan), which has an Animal Welfare Assurance on file (assurance number A5561-01) at the Office of Laboratory Animal Welfare of the National Institutes of Health (NIH, USA). Fish were raised under 12:12 light-dark (L/D) cycles during the first 5 days after birth, unless otherwise stated.

**Transgenic fish lines.** Tg[UAS:mRFP1-TDP-43z] was generated by synthesizing a *Tol2* transposon-based cassette (UAS:mRFP1-TDP-43z) carrying the zebrafish *tardbp* (Genbank accession # NM_2014476) that was tagged with mRFP1 (Genbank accession # AF506027.1) at the N-terminus without linker and placed downstream of x5 upstream activation sequence (UAS)[61] (pBMH-T2ZUASRzT43, Biomatik). For the construction of Tg[UAS:mRFP1-CRY2olig], the zebrafish codon-optimized photolyase homology region (PHR) of *Arabidopsis thaliana* CRY2 carrying the E490G mutation (CRY2olig)[34] was synthesized (Biomatik) and N-terminally tagged with mRFP1 with a linker peptide TRDISIE encoded by ACG CGT GAT ATC TCG ATC GAG (mRFP1-CRY2olig). The mRFP1-CRY2olig fragment was fused to 5xUAS, cloned into the Tol2-transposon cassette. For the construction of Tg[UAS:opTDP-43z], the mRFP1-TDP-43z fragment was C-terminally fused to CRY2olig with the linker peptide TRDISIE (opTDP-43z). The opTDP-43z fragment was fused to 5xUAS, cloned into the Tol2-transposon cassette. NEBuilder HiFi DNA Assembly Master Mix was used for the vector construction. For the generation of Tg[mnr2b-hs:EGFP-TDP-43z], EGFP was directly fused to zebrafish *tardbp* gene (EGFP-TDP-43z) and the resulting EGFP-TDP-43z was linked to the *hsp70l* promoter (650 bp) and introduced into downstream of the *mnr2b* 5′UTR in the *mnr2b*-BAC DNA (CH211-172N16, BACPAC Resources Center) via homologous recombination using a Km$^r$-resistance as essentially described[62]. Tg[mnr2b-hs:opTDP-43h] and Tg[mnr2b-hs:opTDP-43h$^{A315T}$] lines were generated by the same procedure except that opTDP-43h and opTDP-43h$^{A315T}$ were used, respectively, instead of EGFP-TDP-43z. The opTDP-43h consists of the zebrafish-codon-optimized human TDP-43 that is fused directly to the zebrafish-codon-optimized mRFP1 at the N-terminus and indirectly to CRY2olig via the linker peptide TRDISIE. opTDP-43h$^{A315T}$ is identical to opTDP-43h except the A315T mutation (GCT>ACT). All transgenic lines were created via *Tol2*-mediated transgenesis.

**Visualization of the nucleus with fluorescent H2afva.** The open reading frame of *h2afva* encoding a histone H2A variant (NM_153644.1) was C-terminally tagged with EGFP or mRFP1 with a linker nucleotide (GGA GGC TCG AAT CTC GAG) and cloned into the UAS vector pT2MUASMCS[33] to generate pUAS:h2afva-EGFP and pUAS:h2afva-mRFP1, respectively. Twenty-five pg of these plasmids were injected into the embryos at the one-cell stage.

**TDP-43 knockout.** For the generation of *tardbp* and *tardbpl* knockout fish, target sequences for Cas9-mediated cleavage were searched by CRISPRscan[63]. The target sequences CAAGACTTAAAAGACTACTTcgg and CAAGACTTAAAA-GACTACTTcgg, where the protospacer adjacent motifs (PAMs) are indicated by lower cases, were chosen for the generation of *tardbp-n115* and *tardbpl-n94* alleles, respectively. hSpCas9 was in vitro-transcribed with mMESSAGE mMACHINE Kit (Thermo Fisher Scientific, AM1340) by using pCS2 + hSpCas9 plasmid as a template (a gift from Masato Kinoshita, Addgene plasmid # 51815). Wild type embryos were injected with 25 pg of sgRNA and 300 pg of *hSpCas9* mRNA at the one-cell stage.

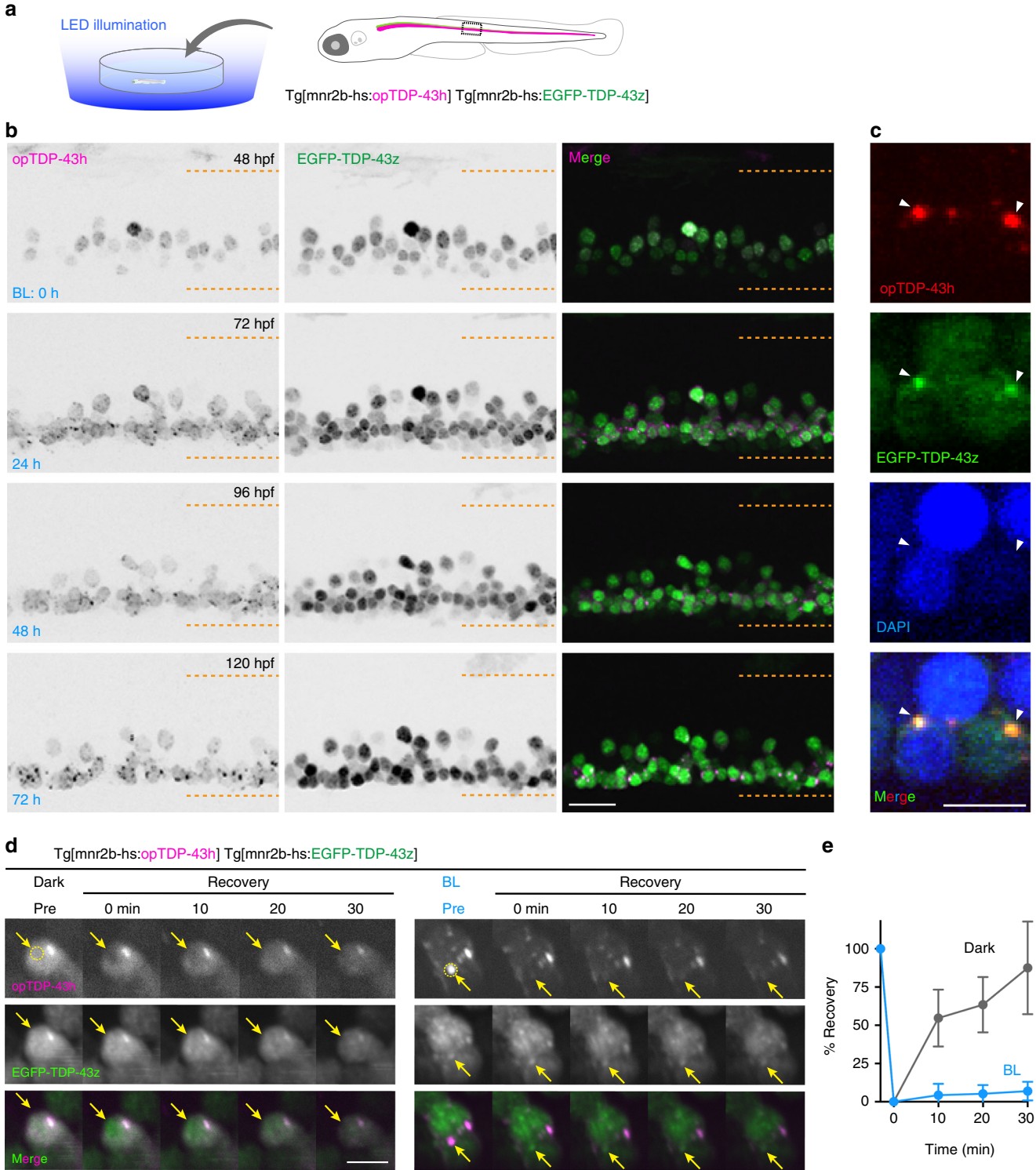

**Fig. 6 Long-term illumination induces opTDP-43h aggregation that seeds non-optogenetic TDP-43 aggregation. a** Chronic field illumination of unrestrained Tg[mnr2b-hs:EGFP-TDP43z] Tg[mnr2b-hs:opTDP-43h] fish by blue LED light. **b** Live imaging of the spinal motor column from 48 to 120 hpf. Horizontal dashed lines demarcate approximate positions of dorsal and ventral limits of the spinal cord. **c** Cytoplasmic opTDP-43h foci colocalize with EGFP-TDP-43z. **d** FRAP analyses of nuclear opTDP43h that had not been exposed to blue light (left, Dark Pre) and cytoplasmic opTDP43h foci that had been induced by a 72-h blue light illumination (right, BL Pre). Bleaching was performed at 120 hpf. Yellow dashed circles (Pre) include photobleached area and arrows indicate the bleached position. **e** Quantification of fluorescent recovery. The results were obtained from 6 cells in independent 6 animals for each condition. Error bars show SD, and the center of the error bars is the average value. The bars indicate 20 µm (**b**), 5 µm (**c, d**). Error bars show SD.

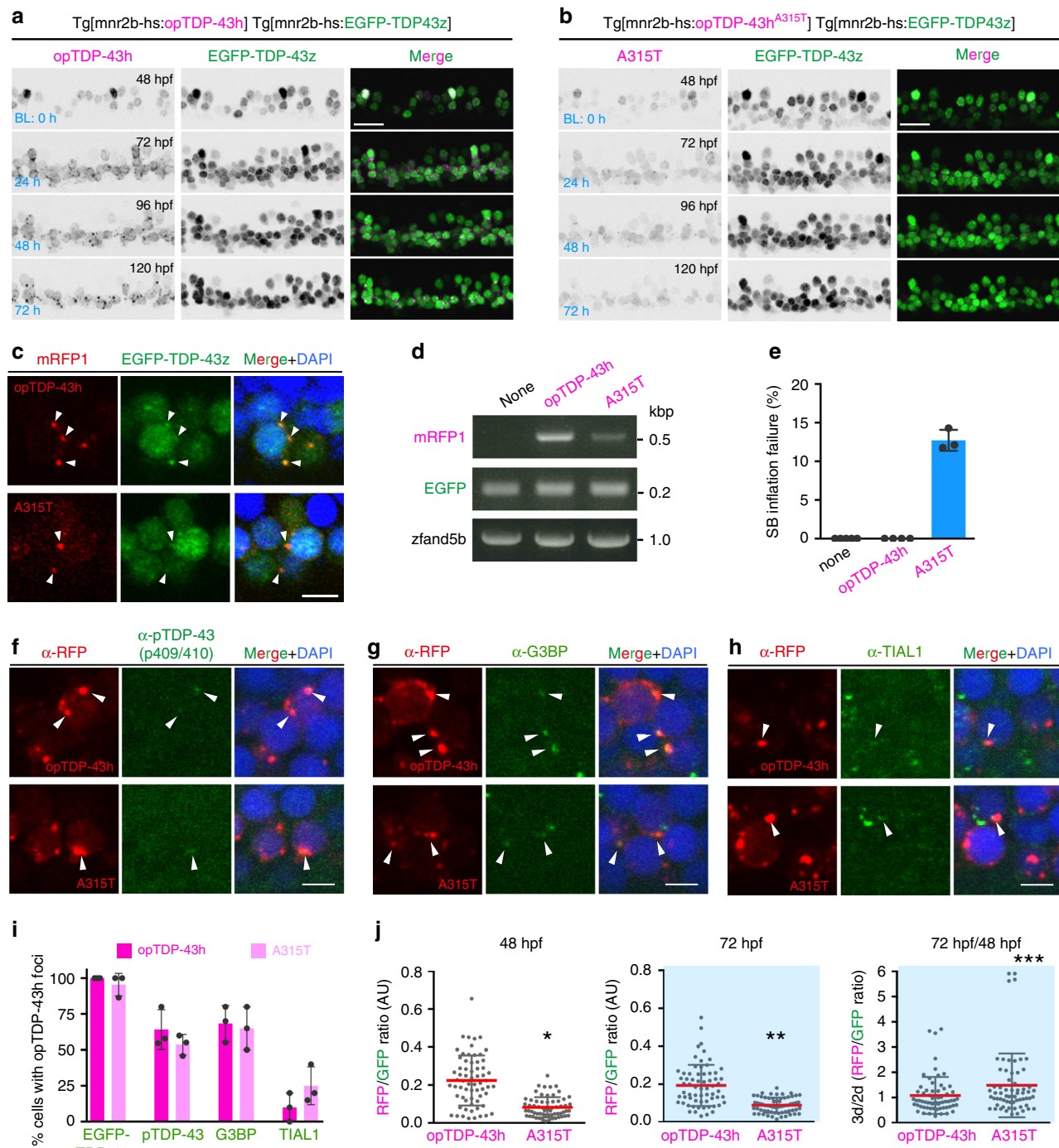

**Fig. 7 The IDR mutation A315T enhances protein stability and oligomerization-dependent toxicity. a, b** Live imaging of the spinal motor column from 48 to 120 hpf. **c** Chronically light-stimulated opTDP-43h (top) and opTDP-43h$^{A315T}$ (bottom) aggregate in the cytoplasm and seed EGFP-TDP-43z aggregation. Arrowheads indicate opTDP-43 and opTDP-43h$^{A315T}$ aggregates that contain EGFP-TDP-43z. **d** RT-PCR analysis for opTDP-43h and opTDP-43h$^{A315T}$ transcripts at 72 hpf. **e** Failure rate of swimming bladder (SB) inflation of Tg[mnr2b-hs:EGFP-TDP43z] (none), Tg[mnr2b-hs:EGFP-TDP43z] Tg[mnr2b-hs:opTDP-43h], and Tg[mnr2b-hs:EGFP-TDP43z] Tg[mnr2b-hs:opTDP-43h$^{A315T}$] (A315T) larvae at 120–144 hpf. The average failure rates were defined from at least three independent assays where six or more fish were illuminated (Source data are provided as a Source Data file). SB inflation failure was not observed when fish were raised under normal dark light cycles ($N > 100$ for each). **f–i** Immunofluorescence analyses of phospo-TDP-43 (**f**), G3BP (**g**), TIAL1 (**h**) against cytoplasmic opTDP-43h and opTDP-43h$^{A315T}$ aggregates at 120 hpf. At least, twenty cells with distinct opTDP-43h or opTDP-43h$^{A315T}$ aggregates were examined for each of three independent fish. **j** Stability of opTDP-43h or opTDP-43h$^{A315T}$. Fluorescence intensities of opTDP-43h or opTDP-43h$^{A315T}$ (RFP) relative to EGFP-TDP-43z (GFP) were examined at 48 hpf and 72 hpf, in the same set of 64 cells from three independent animals. *$p < 0.0001$, **$p < 0.0001$, ***$p = 0.03$ (unpaired t-test, two-tailed). The bars indicate 20 μm (**a, b**) and 5 μm (**c, f–h**). Error bars show SD.

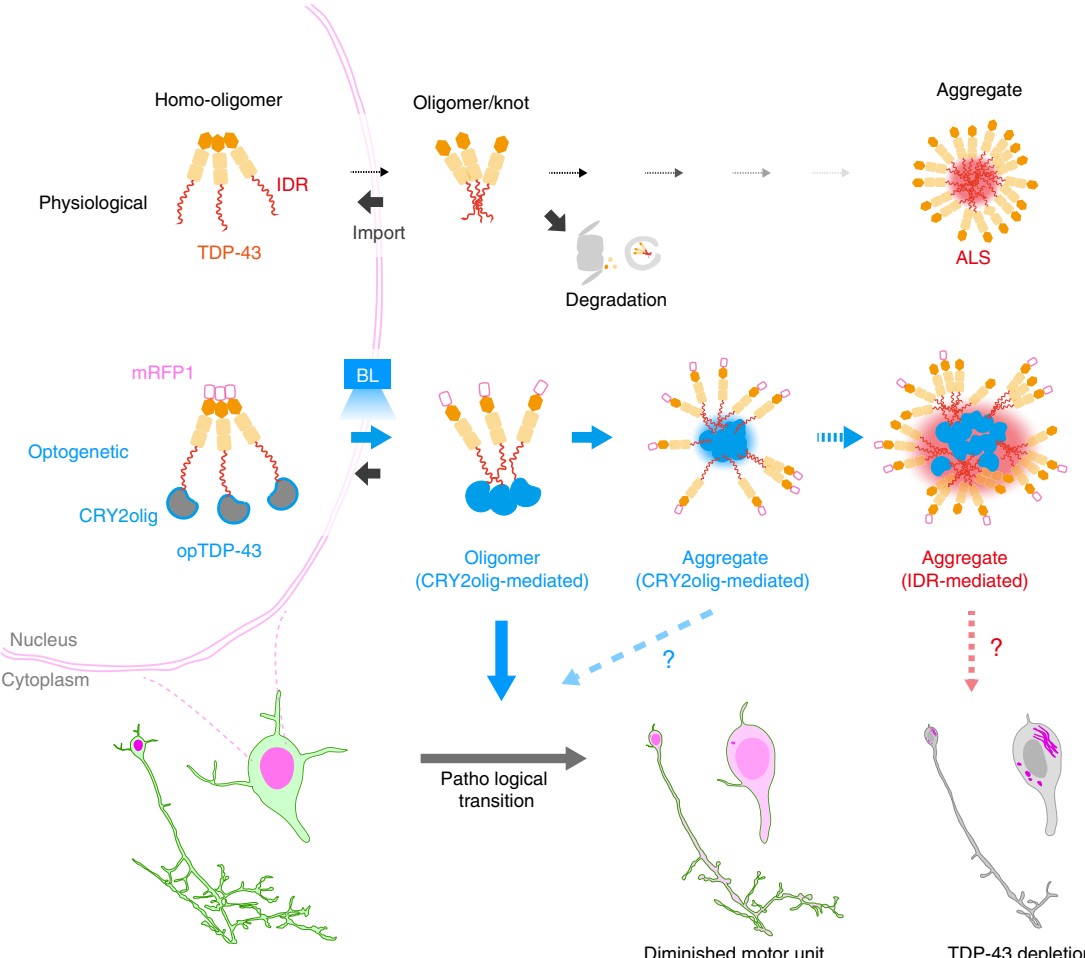

**Fig. 8 Sequentially regulated illumination-triggered TDP-43 knot and aggregate formation.** In physiological conditions, TDP-43 forms oligomers via its N-terminus and is primarily localized in the nucleus. Spinal motor neurons keep the cytoplasmic concentration of TDP-43 oligomers at a low level to prevent them from turning into toxic irreversible oligomers mediated by the C-terminus IDRs (toxic "knots"), which possess competence for developing into pathological TDP-43 aggregates, a hallmark of ALS. CRY2olig-driven opTDP-43 oligomerization promotes pathological change of the motor neurons, such as axon retraction associated with myofiber denervation, prior to accumulation of distinct cytoplasmic aggregates. Whether CRY2olig-diriven opTDP-43 aggregates are toxic to motor neurons and whether CRY2olig-driven aggregates eventually deplete endogenous nuclear TDP-43 pools are unknown.

**Rescue of TDP-43 knockout fish via mRNA injection.** For the expression of human and zebrafish TDP-43 and its derivatives via mRNA injection, the open reading frames of zebrafish *tardbp* (TDP-43z), zebrafish-codon optimized human TDP-43 (TDP-43h), mRFP1-tagged zebrafish *tardbp* (mRFP1-TDP-43z) and opTDP-43z was cloned into pCS2+ vector in vitro-transcribed with mMESSAGE mMACHINE Kit. First, we injected varied amount of TDP-43z mRNA into the offspring obtained from incrosses of parental zebrafish carrying homozygous *tardbp-n115* and heterozygous *tardbpl-n94* mutation or heterozygous *tardbp-n115* and homozygous *tardbpl-n94* mutation at the one cell stage. After investigating the presence or absence of blood flow at 36–48 hpf, all fish were subjected individually to genotyping for *tardbp-n115* and *tardbpl-n94* alleles. The uninjected *tardbp-n115 tardbpl-n94* double homozygotes displayed a swollen heart that was beating, but the blood cells were completely stacked on the yolk surface and cannot reach the heart. An injection of 300 ng of TDP-43z mRNA was the most effect effective in restoring blood flow (up to 40% of the double homozygotes) of the double homozygotes with a minimum developmental abnormality due to overexpression. Throughout the assay, we scored that the blood flow was "rescued" when any blood cell flowing through the beating heart was observed. The function of TDP-43h and TDP-43 derivatives were tested by the microinjection of 300 ng mRNA each. The *tardbpl-n115* allele was identified by performing Heteroduplex Mobility Assay (HMA) against PCR product obtained with a primer pair: tardbpl-6F3 (5′-gcc aga taa taa gag gaa gat gga-3′) and tardbpl-6R3 (5′-tga cag tac aaa gac aaa cac cac-3′). The *tardbpl-n94* allele was similarity identified by using a primer pair: tardbpl-4F2 (5′-caa tca ctg aat gaa tgc act ttt-3′) and tardbpl-4R2 (5′-gtt tgc tta tac taa cct gca cca-3′).

**Blue light illumination.** Short-term (<4 h) light stimulations of mRFP1-CRY2olig and opTDP-43z were carried out by embedding fish in the 0.8–1% low-melting

agarose (NuSieve® GTG® Agarose, Lonza) and conducting confocal scanning with the laser with 473 nm wave length using an Olympus FV1200 microscope. The average optical power of the confocal laser was ~44.66 μW per cm². For longer-term illumination, fish were raised in 6-well dish (FALCON, 353046) with 8 ml E3 buffer, and the dish was placed on a blue LED panel. The intensity of blue LED light that reached the E3 buffer was ~0.69 mW per cm² and its wavelength peaked at 456 nm.

**Immunofluorescence.** Zebrafish larvae were fixed for overnight in PBT (Phosphate Buffered Saline, pH 7.4, with 0.25% Triton X-100) containing 4% paraformaldehyde (#15710, Electron Microscopy Sciences). After wash with PBT, fish were incubated at 70 °C in 150 mM Tris-HCl, pH 9) for 15 min, permeabilized in PBT with 0.025% trypsin and 0.01% EDTA for an hour on ice and treated with antibodies after an hour of blocking with PBT with 1% bovine serum albumin, 2% normal goat serum and 1% dimethyl sulfoxide. For the mono- and poly-ubiquitinated protein staining, Tg[SAGFF73A] Tg[UAS:opTDP-43z] fish at 31.5 hpf that had illuminated with a blue light were taken out from the agarose and immediately subjected to immunofluorescence. The mouse monoclonal antibody for mono- and polyubiquitinate conjugates (FK2, Enzo, 1:100) and goat anti-mouse IgG Alexa Fluor 488 (Molecular Probes, 1:1000) were used as primary and secondary antibodies, respectively. For the detection of opTDP-43z and opTDP-43h, the rabbit anti-RFP polyclonal antibody (pAb, MBL, 1:100) and goat anti-rabbit IgG Alexa Fluor 633 (Molecular Probes, 1:1000) were used as primary and secondary antibodies, respectively. Mouse anti-phospho TDP-43 (pS409/410) (TIP-PTD-MO1, Cosmo Bio, 1:100) and mouse anti-human G3BP antibody (611127, BD Transduction Laboratories, 1:100) were detcted by goat anti-mouse IgG Alexa Fluor 488 (Molecular Probes, 1:1000). Rabbit anti-TIAL1 antibody (NBP1-79932,

Novus Biologicals, 1:100) was detected by goat anti-rabbit IgG Alexa Fluor 488 (Molecular Probes, 1:1000).

**Microscopy.** A fluorescence stereomicroscope (MZ16FA, Leica) equipped with a CCD camera (DFC300FX, Leica) was used to observe the whole fish (Supplementary Figs. 1d, 2). Fixed fish was examined on Olympus FV1200 laser confocal microscope with 40x silicone immersion objectives (Figs. 2f, 6c, 7c, 7f–h, Supplementary Fig. 6). All other images were acquired from live fish embedded in 0.8–1% low-melting agarose (NuSieve® GTG® Agarose, Lonza) on a Glass Base dish (IWAKI, 3010-035) with Olympus FV1200 laser confocal microscope with ×20 water immersion objective (NA1.0). For confocal imaging, fish were raised in embryonic buffer containing 0.003% (w/v) N-Phenylthiourea (SIGMA, P7629) to inhibit melanogenesis. Confocal images were acquired as serial sections along the z-axis and analyzed with Olympus Fluoview Ver2.1b Viewer and Image J, and processed for presentation with Adobe Photoshop CS6. The axon length and branching frequency were measured by Imaris Filament Tracer. Morphological analyses of CaPs were restricted to the spinal segment 14–17 before 50 hpf and to 13–17 during 56–72 hpf. A neurite with more than 5 μm of length was counted as branch.

**Fluorescence recovery after photobleach (FRAP) analysis.** FRAP experiments were performed on Olympus FV1200 laser confocal microscope with ×20 water immersion objective (NA1.0). Photobleaching of opTDP-43h was conducted by scanning of a region of interest (ROI) set in the nucleus or cytoplasm, which was determined using EGFP-TDP-43z signal as a reference, with 599-nm laser at 100% intensity for 5 s. Fluorescence recovery was monitored at 10 min intervals for 30 min. For nuclear opTDP-43h bleaching (Dark), the shape of the nucleus was determined based on EGFP-TDP-43z signal and the opTDP-43h signal included in the nuclear ROI was used to estimate the photobleaching due to the post-bleach imaging for fluorescence recovery. Similarly, for cytoplasmic opTDP-43h foci bleaching (BL), cells with at least two distant cytoplasmic foci were chosen, and the one that was not beached was used to estimate the photobleaching during the post-bleach imaging for fluorescence recovery. The recovery of opTDP-43h signal was determined by subtracting the signal reduction caused by the post-bleaching imaging.

**RT-PCR.** The total RNA was prepared from Tg[mnr2b-hs:EGFP-TDP43z] (none), Tg[mnr2b-hs:EGFP-TDP43z] Tg[mnr2b-hs:opTDP-43h] (opTDP-43), and Tg [mnr2b-hs:EGFP-TDP43z] Tg[mnr2b-hs:opTDP-43h$^{A315T}$] (A315T) larvae at 72 hpf (17 larvae each) by homogenizing in 1 ml of Trizol Reagent (Life Technologies). Three μg of the total RNA is used for cDNA synthesis using oligo dT (SuperScriptn®III First-Strand, Invitrogen). opTDP-43h and opTDP-43h$^{A315T}$ were detected by a primer pair against the zebrafish codon-optimized mRFP1: zmRFP1-123f (5′-TCA CAG AGC TAA ACT GAA GGT CAC-3′) and zmRFP1-633r (5′-GAC GAT GGT ATA GTC TTC GTT GTG-3′). EGFP-TDP-43z was detected by a primer: EGFP-f2s (5′-CAC ATG AAG CAG CAC GAC TTC T-3′) and EGFP-r5s (5′-ACG TTG TGG CTG TTG TAG TTG T-3′). zfand5b expression was detected by a primer pair: zfand5b–133f (5′-ATA GTA CAC ACC GAA ACG GAC AC-3′) and zfand5b-772r (5′-TTA TAT TCT CTG GAT TTT ATC GGC-3′)

**Quantification of stability of opTDP-43h and opTDP-43h$^{A315T}$.** The z-stack images of the spinal motor column were first created by Sum Slices function of Image J. Then, the ROIs covering the somas of individual spinal motor neurons were determined based on the contour of weak cytosolic fluorescence of EGFP-TDP-43z that became clearly visible by signal enhancement. Fluorescence intensities of opTDP-43h, opTDP-43h$^{A315T}$, and EGFP-TDP-43z in the individual ROI were measured by Image J software.

**Statistics and reproducibility.** Statistical analyses were performed using Graph-Pad Prism Software. Micrographs shown are the representative of results from at least three independent animals (except in Fig. 5e, mRFP1-CRY2olig, 2 animals) in at least two independent experiments, which gave similar results, except that the cDNA samples collected in one experiment was used for RT-PCR for Fig. 7d. Sample sizes are reported in figure legends.

**Reporting summary.** Further information on research design is available in the Nature Research Reporting Summary linked to this article.

## Data availability
The data that support the findings in this study are available within the article and its Supplementary Information file, and from KA (kasakawa@nig.ac.jp) on reasonable request. The source data underlying Figs. 1f, g, 2d, 4b, 5c, d, 6e, 7d, e, j, Supplementary Figs. 1e and 6 are provided as a Source Data file.

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

## Acknowledgements

The authors would like to thank Drs Keiko Imamura, Haruhisa Inoue, Shin Kwak for valuable discussions and Kawakami lab members for generous support. This work was supported by SERIKA FUND (K.A.), The Nakabayashi Trust For ALS Research (K.A.), THE KATO MEMORIAL TRUST FOR NAMBYO RESEARCH (K.A.), Daiichi-Sankyo Foundation of Life Science (K.A.), Takeda Science Foundation (K.A.), KAKENHI Grant numbers JP16K07045 (K.A.), JP19K06933 (K.A.), National BioResource Project from Japan Agency for Medical Research and Development (AMED) (K.K.), and KAKENHI Grant Numbers JP15H02370 (K.K.).

## Author contributions

K.A. conceived the research, designed and performed the experiments. K.A. and K.K. analyzed the data and wrote the manuscript with inputs from H.H.

## Competing interests

A patent application covering this work (application no. JP2018-186569, PCT/JP2019/037829) has been filed in which Inter-University Research Institute Corporation Research Organization of Information and Systems, and K.A. and K.K. are the inventors.
