## [Peer Review File · Nature Communications]

Reviewers' comments:

Reviewer #1 (Remarks to the Author):

In this well-written study, the authors established an elegant in vivo system to study TDP-43 toxicity in zebrafish motor neurons. They used light to induce the translocation of ectopically overexpressed TDP-43 to the cytoplasm, where toxic aggregates form in a time-dependent fashion. Although similar optogenetic approaches have been published recently (Zhang et al., *Elife* 2019; Mann et al., *Neuron* 2019), the current study is a nice contribution to the field, and their unique in vivo system in transparent zebrafish will be useful for addressing a number of interesting questions in the future. Here, they concluded that (1) cytoplasmic TDP-43 without aggregate formation can cause neuronal defects; (2) the long-term presence of TDP-43 in the cytoplasm can lead to time-dependent aggregate formation; (3) cytoplasmic TDP-43 aggregates recruit endogenous TDP-43; (4) the intrinsically disordered region of TDP-43 mediates TDP-43 oligomerization, a process that is enhanced by an ALS-associated mutation. Although several of these results are not entirely conceptually novel, they provide further evidence for the hypothesis that the formation of cytoplasmic TDP-43 aggregates can be separated into different phases and that different forms of cytoplasmic TDP-43 may cause toxicity through different mechanisms.

1. A major concern is the statistical analysis in Figure 2D and 2E. Are the values at other time points (150, 180, 210 min) statistically significant? The conclusions in the text that “the cytoplasmic opTDP-43z gradually increased” and “the nuclear opTDP-43z signal decreased slightly but significantly over time during the illumination” need to be supported by solid statistical analysis.

2. Scale bars should be added to Figures 1A, 2C, 3C, 4K, and 5I.

3. Please delete the actual p value from Figure 4D to keep it consistent with other figures.

4. In Figures 2D and E, 5C–E, 6I, and S1E and some other panels, the x-axis is not visible.

Reviewer #2 (Remarks to the Author):

In the manuscript entitled “Optogenetic modulation of TDP-43 oligomerization fast-forwards ALS-related pathologies in the spinal motor neurons”, the authors generate the first in vivo animal system for optogenetic spatial and temporal control of light-induced TDP-43 oligomerization. This system would allow the authors to perform precise experiments and determine molecular mechanisms that drive TDP-43 aggregation and pathology in vivo. However, the authors provide only a rudimentary and tantalizing characterization of the system and fail to use the power of the optogenetics to ask these most interesting questions that could move the neurodegeneration field forward. Furthermore, many of the experiments and analysis given require clarification and revised analysis.

Major critiques organized by figure:

Figure 1. In Fig. 1C, the reader would be better able to evaluate the data if both channels were shown individually

Figure 2. In Fig. 2, the data presented for oligomerization of mRFP1-CRY2olig (control) and opTDP-43z are not even directly comparable. First, it is unclear why different drivers are used for the control mRFP-CRY2olig vs mRFP-tardbp-CRY2olig [TgSAIG213A] vs [SAGFF73A]. In addition, different cell types and different time scales for blue light stimulation are used. This figure does not allow for satisfying comparison of control mRFP1-CRY2olig (control) and opTDP-43z oligomerization.

Figure 3. In this figure, the authors state that TDP-43 does not form oligomers in spinal motor neurons (Fig. 3B) or sensory neurons (Fig. 3C), but instead TDP-43 is mislocalized from the nucleus to the cytoplasm. The data presented, however, is not convincing for several reasons. First, it is unclear whether the authors have accounted for photobleaching in their analysis; loss of signal due to bleaching over several hours of blue light stimulation may contribute to loss of nuclear signal. There is no mention of photobleach correction in the methods. TDP-43 nuclear:cytoplasmic intensity ratio with photobleach correction would be a far better method of analysis. Finally, as mentioned above, the reader would be better able to evaluate the data if both channels were shown individually. Please also include images of control animals expressing mRFP1-CRY2olig for comparison to mRFP-opTDP-43z.

Biochemical analysis of the spinal cord with nuclear and cytoplasmic fractionation and western blot that demonstrates a shift of TDP-43 from the nucleus to the cytoplasm upon blue light stimulation would also be more convincing. Is endogenous TDP-43 recruited to the cytoplasm?

Figure 4. In this figure, the authors address (1) how optogenetic mislocalization of TDP-43 affects axon outgrowth and branching and (2) whether endogenous TDP-43 get recruited to the cytoplasm. The authors use another transgene to visualize the non-optogenetic pool of TDP-43 (which is not required to the cytoplasm), but can immunofluorescence be performed to visualize endogenous TDP-43? Would longer periods of blue light exposure cause mislocalization of non-optogenetic TDP-43 (as in later figure, Fig. 6)?

Figure 5. Why is statistical analysis is not provided for 5E? Quantification of 5H, 5I to show that % of co-localization between VGlut and chrnd reduces more with induction of opTDP-43. It appears that the control animals shown in 5H were not exposed to blue light; how do we know that any differences in VGlut and chrnd colocalization is not simply due to toxicity from blue light stimulation? Appropriate control would be animals expressing CRY2olig that were also exposed to blue light.

Figure 6. Overall, the manuscript text describing this figure was confusing, and perhaps this was due to incorrect figure references (e.g. there is no reference to Fig. 6D in the manuscript text)? In contrast to data presented earlier, the authors show longer periods of light stimulation are able to induce TDP-43 puncta in the cytoplasm in motor neurons! Furthermore, this stimulation protocol was able to recruit non-optogenetic TDP-43 to the cytoplasm. It is unclear why this optimized approach was not used for the earlier experiments. (?)

Finally, the authors also show data using TDP-43 A315T mutant, as a way of disrupting the IDR. However, nearly all disease mutations cluster in the IDR, so why did the authors use this particular mutant? Clarification of the rationale would make this section more satisfying.

Minor critiques:

A few typographical errors are noted. For example, page 6, line 140 of the manuscript, "accessed" should be assessed.

Overall, the manuscript requires major revision and additional mechanistic experiments (and/or potentially a screen, as the authors suggest in the Discussion), before it could be considered for publication in Nature Communications.

Reviewer #3 (Remarks to the Author):

The manuscript by Asakawa and colleagues describe development and characterization of an optogenetic TDP-43 (opTDP-43) model where they were able to regulate TDP-43 aggregation

through exposure of external light in vivo. The authors used elegant zebrafish neuromuscular system to demonstrate that short-term light stimulation reversibly induces cytoplasmic opTDP-43 mislocalization in the spinal motor neurons and axon outgrowth defects. Interestingly, long-term light illumination promotes opTDP-43 forms pathological aggregates in the cytoplasm which recruits non-optogenetic TDP-43 aggregation. There are few cell-based opto models that has been developed and characterized recently but the authors provide first in vivo model system for manipulating TDP-43 in a whole animal model organism. Overall, this is an interesting study that is likely to advance our knowledge about the basic biology of TDP-43 protein in human neurodegenerative diseases.

The authors should address the following issues with clarifications and experimental evidences.

1. The author should provide images showing that WT Tardbp and mRFP1-TDP-43z rescuing the blood circulation phenotypes in the TDP-43 DKP embryos.
2. Is the UAS::mRFP1-Cry2olig (Fig 2B) toxic or have any effect on axonal length upon illumination? The authors should provide images and quantification data showing its effect on axonal length.
3. The authors should also provide images showing opTDP-43z rescuing the blood circulation defect of TDP43DKO embryo to support Sup Figure 1E
4. A control of mRFP1-CRYolig in skeletal muscle at 0 min and 210 min is missing and it should be provided.
5. There is no Figure 4F. The authors should include it since it is discussed on page 8.
6. The sentence in line 253-255 should be restructure. It is an incomplete sentence.
7. Previous studies have shown that pathological aggregates of TDP-43 are phosphorylated. It is not clear if the authors have tested this possibility. It would be great if they could include data showing phosphorylation status of the opTDP-43h foci.
8. Most cases of TDP-43 aggregates associates with stress granules (SGs). Does the opTDP-43h and opTDP-43hA315T associates with any SG markers? The authors should provide data showing optoTDP-43 (WT and mutant) with SG markers.
9. It is not clear that why the authors kept on using the term TDP-43 aggregates. They should perform FRAP analysis to prove that these are aggregates. Otherwise, they should use the term puncta or foci. Little more clarification would have been helpful.

There are several minor issues with writing and the authors should go over the manuscript carefully and fix it.

Response to reviewers' comments

<Reviewers comments were *italicized*. >

Reviewer #1 (Remarks to the Author):

In this well-written study, the authors established an elegant in vivo system to study TDP-43 toxicity in zebrafish motor neurons. They used light to induce the translocation of ectopically overexpressed TDP-43 to the cytoplasm, where toxic aggregates form in a time-dependent fashion. Although similar optogenetic approaches have been published recently (Zhang et al., Elife 2019; Mann et al., Neuron 2019), the current study is a nice contribution to the field, and their unique in vivo system in transparent zebrafish will be useful for addressing a number of interesting questions in the future. Here, they concluded that (1) cytoplasmic TDP-43 without aggregate formation can cause neuronal defects; (2) the long-term presence of TDP-43 in the cytoplasm can lead to time-dependent aggregate formation; (3) cytoplasmic TDP-43 aggregates recruit endogenous TDP-43; (4) the intrinsically disordered region of TDP-43 mediates TDP-43 oligomerization, a process that is enhanced by an ALS-associated mutation. Although several of these results are not entirely conceptually novel, they provide further evidence for the hypothesis that the formation of cytoplasmic TDP-43 aggregates can be separated into different phases and that different forms of cytoplasmic TDP-43 may cause toxicity through different mechanisms.

Answer:

We greatly appreciate the positive comments from the reviewer #1.

1. A major concern is the statistical analysis in Figure 2D and 2E. Are the values at other time points (150, 180, 210 min) statistically significant? The conclusions in the text that “the cytoplasmic opTDP-43z gradually increased” and “the nuclear opTDP-43z signal decreased slightly but

significantly over time during the illumination” need to be supported by solid statistical analysis.

Answer:

We analyzed the data and confirmed that the increase of cytoplasmic opTDP-43z and decrease of nuclear opTDP-43z were statistically significant after 150 min and 120 min, respectively. The p values were shown in the figure legend (line 859).

2. Scale bars should be added to Figures 1A, 2C, 3C, 4K, and 5I.

Answer:

We added the scale bars to these figures.

3. Please delete the actual p value from Figure 4D to keep it consistent with other figures.

Answer:

The p value was deleted.

4. In Figures 2D and E, 5C-E, 6I, and S1E and some other panels, the x-axis is not visible.

Answer:

We checked the figures and added the x axes, where absent.

Reviewer #2 (Remarks to the Author):

In the manuscript entitled “ Optogenetic modulation of TDP-43 oligomerization fast-forwards ALS-related pathologies in the spinal motor neurons” , the authors generate the first in vivo animal system for optogenetic spatial and temporal control of light-induced TDP-43

oligomerization. This system would allow the authors to perform precise experiments and determine molecular mechanisms that drive TDP-43 aggregation and pathology in vivo. However, the authors provide only a rudimentary and tantalizing characterization of the system and fail to use the power of the optogenetics to ask these most interesting questions that could move the neurodegeneration field forward. Furthermore, many of the experiments and analysis given require clarification and revised analysis.

Major critiques organized by figure:

Figure 1. In Fig. 1C, the reader would be better able to evaluate the data if both channels were shown individually

Answer:

As suggested, we separated the green and magenta channels so that the cell morphology and opTDP-43 could be independently visualized.

Figure 2. In Fig. 2, the data presented for oligomerization of mRFP1-CRY2olig (control) and opTDP-43z are not even directly comparable. First, it is unclear why different drivers are used for the control mRFP-CRY2olig vs mRFP-tardbp-CRY2olig [TgSAIG213A] vs [SAGFF73A]. In addition, different cell types and different time scales for blue light stimulation are used. This figure does not allow for satisfying comparison of control mRFP1-CRY2olig (control) and opTDP-43z oligomerization.

Answer:

We agree with the reviewer #2 that the original Figure 2 does not allow comparison of oligomerization capacities between mRFP1-CRY2olig and opTDP-43z. In the revised manuscript, we used the same Gal4 driver Tg[SAGFF73A] and focused on the same cell type (skeletal muscle cells). The results are shown in Figure 2B and C. As the mRFP1-CRY2olig clustering in the spinal motor neurons in the original Figure 2 does not fit into the revised Figure 2, we described it in Sup. Figure 3.

Figure 3. In this figure, the authors state that TDP-43 does not form oligomers in spinal motor neurons (Fig. 3B) or sensory neurons (Fig. 3C), but instead TDP-43 is mislocalized from the nucleus to the cytoplasm. The data presented, however, is not convincing for several reasons. First, it is unclear whether the authors have accounted for photobleaching in their analysis; loss of signal due to bleaching over several hours of blue light stimulation may contribute to loss of nuclear signal. There is no mention of photobleach correction in the methods. TDP-43 nuclear:cytoplasmic intensity ratio with photobleach correction would be a far better method of analysis.

Answer:

In this figure, we would like to conclude that opTDP-43z does not form distinct cytoplasmic foci in the spinal motor neurons and sensory neurons after 270 min illumination, but instead opTDP-43z mislocalizes to the cytoplasm (we do believe that opTDP-43z forms oligomers).

We agree with the reviewer #2 that consideration of photobleaching is necessary to precisely measure the cytoplasmic mislocalization of opTDP-43z. We found it difficult, however, to estimate the total amount of opTDP-43z within the cell overtime, which is necessary to estimate the extent of photobleaching, because, if cytoplasmic mislocalization happened, opTDP-43z would be dispersed not only in the soma but also throughout the axon, which accounts for a major part of the cytoplasm but cannot be fully covered by confocal imaging. Therefore, in theory, the reduced opTDP-43z signal could be due to either photobleaching or mislocalization to the cytoplasm including axon, or both.

Nonetheless, what we observed in this experiment was the increase in the opTDP-43z signal at the periphery of the cell after the 270 min illumination, which is indicated by blue arrows (Figure 3B and C). We think that the increase itself proves the cytoplasmic opTDP-43z mislocalization, even when the reduction of nuclear opTDP-43z signal was caused by photobleaching. We agree that the previous Figure3 was not convincing in that it was not clearly demonstrated that the increase of opTDP-43z signal

in the cell periphery occurred indeed in the cytoplasm. Therefore, to improve Figure 3, we performed the same set of experiments except that the nuclear was visualized with EGFP-tagged histone H2A variant (h2afva-EGFP) instead of visualizing soma with monomeric EGFP (Figure 3E). By doing so, we found that opTDP-43z colocalized with h2afva-EGFP prior to the illumination, but became also detectable outside of the h2afva-EGFP signal after the blue light illumination. We believe that this demonstrates that opTDP-43 mislocalized to the cytoplasm.

Finally, as mentioned above, the reader would be better able to evaluate the data if both channels were shown individually.

Answer:

As suggested, we showed different colors in different panels.

Please also include images of control animals expressing mRFP1-CRY2olig for comparison to mRFP-opTDP-43z.

Answer:

We performed the experiments for mRFP1-CRY2olig and found that mRFP1-CRY2olig displayed widespread distribution in the soma before illumination, and it immediately formed large foci once illumination began.

Biochemical analysis of the spinal cord with nuclear and cytoplasmic fractionation and western blot that demonstrates a shift of TDP-43 from the nucleus to the cytoplasm upon blue light stimulation would also be more convincing.

Answer:

We thank the reviewer #2 for suggesting alternative approaches to prove the opTDP-43 cytoplasmic mislocalization. Because zebrafish embryos are tiny and the number of motor neurons is too few to perform biochemical analyses after dissociating fish and collecting cells by fluorescence-activated cell

sorting, we are afraid that biochemical fractionation is impractical in this particular case. We also do not know, to our knowledge, any study performing biochemical analyses on neurons with this small number in zebrafish. We hope that the localization analysis of opTDP-43 with h2afva-EGFP (Figure 3E) would suffice to conclude that opTDP-43 mislocalizes to the cytoplasm upon blue light illumination.

Is endogenous TDP-43 recruited to the cytoplasm?

Answer:

Because opTDP-43z contains the entire sequence of zebrafish TDP-43z/Tardbp, opTDP-43z cannot be distinguished from endogenous TDP-43z/Tardbp by immunofluorescence. This is one of the main reasons why we developed EGFP-TDP-43z to analyze how non-optogenetic TDP-43 are affected by opTDP-43z.

Figure 4. In this figure, the authors address (1) how optogenetic mislocalization of TDP-43 affects axon outgrowth and branching and (2) whether endogenous TDP-43 get recruited to the cytoplasm. The authors use another transgene to visualize the non-optogenetic pool of TDP-43 (which is not required to the cytoplasm), but can immunofluorescence be performed to visualize endogenous TDP-43?

Answer:

As explained above, opTDP-43z cannot be distinguished from endogenous TDP-43 by immunofluorescence.

Would longer periods of blue light exposure cause mislocalization of non-optogenetic TDP-43 (as in later figure, Fig. 6)?

Answer:

In this experiment, where fish were embedded in the agarose and scanned with confocal laser for blue light illumination, we avoided illuminating fish

more than 4 hours because prolonged agarose-embedding and light exposure could delay fish growth and sometimes decrease fish viability, making the analysis of physiological consequences of opTDP-43z illumination difficult. In the revised manuscript, we performed additional experiments to test whether a long-term light exposure itself causes cytoplasmic EGFP-TDP-43z mislocalization independently of opTDP-43z. We illuminated EGFP-TDP-43z with blue LED light for 72 hours (as in Figure 6) in the absence of opTDP-43z and presence of a nuclear marker h2afva-mRFP1 as a reference (Sup. Figure 5). We found that EGFP-TDP-43z stayed within the nucleus as judged by colocalization with h2afva-mRFP1. Thus, although unable to directly test the reviewer #2's question by confocal set-up, we think that the prolonged blue light illumination alone neither induced cytoplasmic mislocalization nor foci formation of EGFP-TDP-43z, in the absence of opTDP-43h. We added the sentence describing this result in line 283.

Figure 5. Why is statistical analysis is not provided for 5E?

Answer:

In figure 5E, we describe the fluctuation in the terminal number before and after the illumination by showing what percentages of DCCTs increased or decreased their terminal numbers analyzed in Figure 5C, D). To make this point clearer, we changed the y-axis labeling (Figure 5E).

Quantification of 5H, 5I to show that % of co-localization between VGlut and chrnd reduces more with induction of opTDP-43. It appears that the control animals shown in 5H were not exposed to blue light; how do we know that any differences in VGlut and chrnd colocalization is not simply due to toxicity from blue light stimulation? Appropriate control would be animals expressing CRY2^{olig} that were also exposed to blue light.

Answer:

We thank reviewer#2 for pointing out the importance of quantification of denervation. Unlike the visualization of axonal morphology with monomeric EGFP (as in Figure 5A-E), labeling of presynaptic boutons by VAMP2-Venus (V2V) does not necessarily allow to trace axon morphology due to its punctate nature of staining pattern. This is particularly the case at 72 hpf when the V2V puncta of DCCT and ones belonging to other collaterals intermingle each other, making it difficult to trace in a rigorous way all of individual V2V puncta in a DCCT from 56 hpf to 72 hpf. However, we found it feasible to determine whether the V2V and tdT-chrnd puncta at the end of the axon tip of DCCTs disappeared or not in most, if not all, cases. As shown in Figure 5 H, prior to blue light illumination (at 56 hpf), most of V2V puncta at the axon tips were juxtaposed with tdT-chrnd signal, and the frequency of V2V /tdT-chrnd juxtaposition were not affected by opTDP-43 or mRFP1-CRY2olig expression. After the illumination, DCCTs that expressed opTDP-43z reduced the number of V2V terminal, while such reduction was not evident in DCCTs expressing mRFP1-CRY2olig. Based on these data, we would like to draw a conclusion that that light-stimulation of opTDP-43 reduced the number of DCCT terminals with V2V /tdT-chrnd juxtaposition.

Figure 6. Overall, the manuscript text describing this figure was confusing, and perhaps this was due to incorrect figure references (e.g. there is no reference to Fig. 6D in the manuscript text)?

Answer:

We checked that the each figure has their reference in the text.

In contrast to data presented earlier, the authors show longer periods of light stimulation are able to induce TDP-43 puncta in the cytoplasm in motor neurons! Furthermore, this stimulation protocol was able to recruit non-optogenetic TDP-43 to the cytoplasm. It is unclear why this optimized approach was not used for the earlier experiments. (?)

Answer:

Given the prompt response of mRFP1-CRY2_{olig} against blue light (Figure 2, Sup. Figure 3), we began our analysis of opTDP-43z with confocal laser microscope, because it enabled opTDP-43z characterization at a higher spatiotemporal resolution. This type of detailed analyses is nearly impossible in the LED light illumination protocol, which we used in Figure 6 and 7, as it needs repeated embeddings in and removals from the agarose, which are toxic to fish. The confocal laser stimulation against embedded fish allowed us to find cell-type specific propensity of opTDP-43z aggregation and toxicity of opTDP-43z before forming distinct cytoplasmic aggregates. Therefore, for us, the long-term LED light illumination protocol only became possible and reasonable after the detailed kinematic analysis with the confocal laser stimulation-protocol.

Finally, the authors also show data using TDP-43 A315T mutant, as a way of disrupting the IDR. However, nearly all disease mutations cluster in the IDR, so why did the authors use this particular mutant? Clarification of the rationale would make this section more satisfying.

Answer:

We added a sentence regarding the reason why we chose this mutation (in line 307-308).

Minor critiques:

A few typographical errors are noted. For example, page 6, line 140 of the manuscript, “accessed” should be assessed.

Answer:

We corrected the error accordingly, and tried to minimize typographical errors.

Overall, the manuscript requires major revision and additional mechanistic experiments (and/or potentially a screen, as the authors suggest in the Discussion), before it could be considered for publication in Nature

Communications.

Answer:

We hope that, where experiments are practically possible in zebrafish, we addressed all the questions from the reviewer #2. Also, we have performed additional experiments to obtain mechanistic insight into how cytoplasmic opTDP-43h aggregates assemble, as described in Figure 6 and 7. By performing FRAP, we found that cytoplasmic opTDP-43h foci were immobile aggregate with a very low molecular exchange rate. We also found by immunofluorescence, the opTDP-43h aggregates are heterogenous protein assemblies that included consistently non-optogenetic EGFP-TDP-43 but partially C-terminally phosphorylated TDP-43, G3BP, and TIAL. These results provide *in vivo* evidence for the hypotheses that self-seeding of TDP-43 aggregation can take place independently of pathological phosphorylation at S409/S410 and conventional SG assembly and that the self-seeded TDP-43 aggregates could have multiple fate in the spinal motor neurons. We hope that these provide mechanistic insights into how TDP-43 oligomers develop into pathological aggregates in the spinal motor neurons *in vivo*.

With regard to chemical screening, we think this system will be applicable to high-throughput screening in the future with some improvements, but that is not something we can achieve within the reasonable timeframe during the revision.

Finally, we greatly appreciate the reviewer #2's comments instructing us in detail how this manuscript can be improved. We regret that the originally submitted manuscript might have disappointed the reviewer #2 as it appeared to be "rudimentary and tantalizing characterization". We have to admit that the resolution of our analyses in time and space is inferior to one with cultured cells from the view point of cell biology. However, this is because we have to look into the live motor neurons deep in the spinal cord and, at the same time, fully maintain viability of the animal during the experiment. To our knowledge, no study before this one has achieved time-course observation of morphology and

neuromuscular synapse of single identified spinal motor neurons with temporally-tuned intervention of TDP-43 in a vertebrate model. We would greatly appreciate if this point is also taken into consideration to evaluate this work.

Reviewer #3 (Remarks to the Author):

The manuscript by Asakawa and colleagues describe development and characterization of an optogenetic TDP-43 (opTDP-43) model where they were able to regulate TDP-43 aggregation through exposure of external light in vivo. The authors used elegant zebrafish neuromuscular system to demonstrate that short-term light stimulation reversibly induces cytoplasmic opTDP-43 mislocalization in the spinal motor neurons and axon outgrowth defects. Interestingly, long-term light illumination promotes opTDP-43 forms pathological aggregates in the cytoplasm which recruits non-optogenetic TDP-43 aggregation. There are few cell-based opto models that has been developed and characterized recently but the authors provide first in vivo model system for manipulating TDP-43 in a whole animal model organism. Overall, this is an interesting study that is likely to advance our knowledge about the basic biology of TDP-43 protein in human neurodegenerative diseases.

The authors should address the following issues with clarifications and experimental evidences.

Answer:

We greatly appreciate the positive comments from the reviewer #3.

1. The author should provide images showing that WT Tardbp and mRFP1-TDP-43z rescuing the blood circulation phenotypes in the TDP-43 DKP embryos.

Answer:

We provided a movie showing the restored blood circulation in Sup. Movie 1. We would like to mention that the rescue could not be 100 % even with the wild type *tardbp/TDP-43z* because the ectopic expression of *tardbp/TDP-43z* by mRNA injection at a high concentration causes early developmental defects. Here, we tested several mRNA amounts beforehand and found 300 pg, which we used in this study, is an optimal amount that minimized the early toxicity and maximized the blood circulation rescue.

2. Is the UAS::mRFP1-Cry2olig (Fig 2B) toxic or have any effect on axonal length upon illumination? The authors should provide images and quantification data showing its effect on axonal length.

Answer:

We did not detect any significant effect of *mRFP1-Cry2olig* expression on the total axon length and branching frequency (Figure 4D and E). The typical image of a CaP expressing *mRFP1-Cry2olig* was shown in Figure 4C. In the revised manuscript, the original Figure 2B is shown in Sup. Figure 3.

*3. The authors should also provide images showing *opTDP-43z* rescuing the blood circulation defect of *TDP43DKO* embryo to support Sup Figure 1E*

Answer:

We provided a movie showing the rescue by *opTDP-43z* in Sup. Movie 1.

4. A control of mRFP1-CRYolig in skeletal muscle at 0 min and 210 min is missing and it should be provided.

Answer:

We provided data for mRFP1-CRYolig in skeletal muscle in Figure 2B.

5. There is no Figure 4F. The authors should include it since it is discussed on page 8.

Answer:

We regret this mistake in figure labeling. In the revised manuscript, Figure 4C is to show that CaPs expressing opTDP-43z arborized within their inherent ventral innervation territory.

6. The sentence in line 253-255 should be restructure. It is an incomplete sentence.

Answer:

We rewrite the sentence as shown in line 265.

7. Previous studies have shown that pathological aggregates of TDP-43 are phosphorylated. It is not clear if the authors have tested this possibility. It would be great if they could include data showing phosphorylation status of the opTDP-43h foci.

Answer:

We performed immunofluorescence experiments for phosphorylation of opTDP-43h at S409/S410, as shown in Figure 7F and I.

8. Most cases of TDP-43 aggregates associates with stress granules (SGs). Does the opTDP-43h and opTDP-43hA315T associates with any SG markers? The authors should provide data showing opTDP-43 (WT and mutant) with SG markers.

Answer:

We found that antibodies against human G3BP and TIAL consistently recognized heat-shock induced SGs in zebrafish (Sup. Figure 6). Using these antibodies, we performed immunofluorescence experiments for G3BP and TIAL, and presented the results in Figure 7G, H and I.

9. It is not clear that why the authors kept on using the term TDP-43 aggregates. They should perform FRAP analysis to prove that these are

aggregates. Otherwise, they should use the term puncta or foci. Little more clarification would have been helpful.

Answer:

We agree with reviewer #3 for this point. As suggested, we performed FRAP experiments to explore molecular dynamics of opTDP-43 before and after illumination. Results are presented in Figure 6. Also, we avoided using “aggregates” to describe opTDP-43 in the text before the FRAP experiments.

There are several minor issues with writing and the authors should go over the manuscript carefully and fix it.

Answer:

We tried our best to errors in writing.

REVIEWERS' COMMENTS:

Reviewer #1 (Remarks to the Author):

I have no further comments. Thanks.

Reviewer #2 (Remarks to the Author):

The revised manuscript "Optogenetic modulation of TDP-43 oligomerization fast-forwards ALS-related pathologies in the spinal motor neurons", the authors have presented the first in vivo animal system for optogenetic spatial and temporal control of light-induced TDP-43 oligomerization. This system will allow the authors to perform precise experiments and determine molecular mechanisms that drive TDP-43 aggregation and pathology in vivo.

Thank you for addressing the major critiques raised and for providing clarifications -- this reviewer finds revised figures far more convincing now, especially figures 2, 3, and 5. Addition of the data in the new Fig.7 was also interesting, and provides in vivo evidence that TDP-43 cytoplasmic mislocalized puncta can, at least in part co-localize with stress granule markers.

REVIEWERS' COMMENTS:

Reviewer #1 (Remarks to the Author):

I have no further comments.

Thanks.

Reviewer #2 (Remarks to the Author):

The revised manuscript “ Optogenetic modulation of TDP-43 oligomerization fast-forwards ALS-related pathologies in the spinal motor neurons”, the authors have presented the first in vivo animal system for optogenetic spatial and temporal control of light-induced TDP-43 oligomerization. This system will allow the authors to perform precise experiments and determine molecular mechanisms that drive TDP-43 aggregation and pathology in vivo.

Thank you for addressing the major critiques raised and for providing clarifications -- this reviewer finds revised figures far more convincing now, especially figures 2, 3, and 5. Addition of the data in the new Fig.7 was also interesting, and provides in vivo evidence that TDP-43 cytoplasmic mislocalized puncta can, at least in part co-localize with stress granule markers.